# Signatures of Mottness and Hundness in archetypal correlated metals

Xiaoyu Deng [1,4], Katharina M. Stadler[2,4], Kristjan Haule[1], Andreas Weichselbaum[2,3], Jan von Delft [2] & Gabriel Kotliar[1,3]

Physical properties of multi-orbital materials depend not only on the strength of the effective interactions among the valence electrons but also on their type. Strong correlations are caused by either Mott physics that captures the Coulomb repulsion among charges, or Hund physics that aligns the spins in different orbitals. We identify four energy scales marking the onset and the completion of screening in orbital and spin channels. The differences in these scales, which are manifest in the temperature dependence of the local spectrum and of the charge, spin and orbital susceptibilities, provide clear signatures distinguishing Mott and Hund physics. We illustrate these concepts with realistic studies of two archetypal strongly correlated materials, and corroborate the generality of our conclusions with a model Hamiltonian study.

[1] Department of Physics and Astronomy, Rutgers University, Piscataway, NJ 08854, USA. [2] Physics Department, Arnold Sommerfeld Center for Theoretical Physics and Center for NanoScience, Ludwig-Maximilians-Universitat München, 80333 München, Germany. [3] Condensed Matter Physics and Materials Science Department, Brookhaven National Laboratory, Upton, NY 11973, USA. [4] These authors contributed equally: Xiaoyu Deng, Katharina M. Stadler. Correspondence and requests for materials should be addressed to X.D. (email: xiaoyu.deng@gmail.com)

The excitation spectra and transport properties of transition metal oxides at high energy and/or high temperature are well described in terms of dressed atomic excitations with their characteristic multiplet structure. At very low-energy scales, by contrast, metallic systems are well described in terms of strongly renormalized Landau quasiparticles forming dispersive bands. Describing the evolution of the excitation spectrum as a function of energy scale is a fundamental problem in the theory of strongly correlated materials. Starting with the Fermi liquid quasiparticles at the lowest energy scales, and raising the temperature, one can view this evolution as their gradual undressing. Conversely, starting from the high energy end, one can understand the evolution of the excitation spectrum as the screening of the orbital and spin excitations of atomic states, which gradually bind to give rise to quasiparticles. Here we consider the temperature dependence of this screening process for correlated multi-orbital systems with strong on-site atomic-like interactions, involving both a Coulomb repulsion $U$ and Hund's coupling $J$. The former differentiates between different charge configurations without preference for a given spin or orbital configuration, whereas the latter favors the highest spin state.

It is well known that strong correlation effects can arise due to proximate Mott-insulating states in which strong on-site Coulomb repulsion slows down charge fluctuations or even blocks the charge motion and localizes the electrons[1,2]. However, many materials far away from the Mott-insulating state, notably the $3d$ iron-based superconductors[3,4] and ruthenates[5,6], display strong correlation effects as a result of strong Hund coupling rather than the Hubbard $U$. These so-called Hund metals[3–14] were proposed to be a new type of strongly correlated electron system, characterized by spin–orbital separation[9–11].

The existence of different origins of correlations, Coulomb $U$ or Hund $J$, poses an important questions: what are the defining signatures distinguishing Mott and Hund metals? The goal of this work is to answer this question, by pointing out that Hund and Mott metals differ strikingly in the temperature dependencies of their local correlated spectra and the local susceptibilities describing the charge, spin, and orbital degrees of freedom. These differences reflect two distinct screening routes for how quasiparticles emerge from the atomic degrees of freedom, with spin–orbital separation involved for Hund metals, but not for Mott metals.

In this paper, we provide evidence for the two distinct screening routes by investigating (i) two realistic materials and (ii) a model Hamiltonian. For (i) we consider two archetypal materials with non-degenerate orbitals, the Mott system $V_2O_3$[15–17] and the Hund metal $Sr_2RuO_4$[18]. We compute their properties using density functional theory plus dynamical mean-field theory (DFT + DMFT)[19–21], which has been successfully used to describe the available experimental measurements for $V_2O_3$[22–28] and $Sr_2RuO_4$[5,29–31]. For (ii) we study a three-band Hubbard–Hund model (3HHM) with three degenerate bands hosting two electrons. This 3HHM is the simplest model capable of capturing both Hund and Mott physics and the crossover between them as function of increasing $U$[9,10]. Whereas in ref. [10] we focused on $T = 0$, here we focus on temperature dependence. We study the 3HHM using DMFT and the numerical renormalization group (DMFT + NRG). We accurately determine the location of the Mott transition at zero temperature and show that, provided that $J$ is sizeable, the temperature dependence of physical properties for large $U$ near the Mott transition line qualitatively resembles that of $V_2O_3$, while for small $U$ far from the transition it resembles that of $Sr_2RuO_4$. Therefore, our 3HHM results elucidate the physical origin of the differences between these materials. Indeed, we argue that our findings are applicable to general multi-orbital materials and characteristic of the general phenomenology of Mott and Hund physics, independent of material-dependent details, such as the initial band structure.

## Results

**Overview**. We start with an overview of our most important observations. We first identify four temperature scales, characterizing the onset and completion of screening of the orbital and spin degrees of freedom as the temperature is lowered. The scales for the onset of screening, $T_{orb}^{onset}$ and $T_{spin}^{onset}$, are defined as the temperatures at which the static local orbital and spin susceptibilities, $\chi_{orb}$ and $\chi_{spin}$, first show deviations from the Curie behavior, $\chi \propto 1/T$, shown by free local moments. The scales for the completion of screening, $T_{orb}^{cmp}$ and $T_{spin}^{cmp}$, mark the transition of these susceptibilities to Pauli behavior, saturating to constants at very low temperatures. For orientation, Fig. 1 summarizes the behavior of these scales with increasing Coulomb interaction at fixed Hund's coupling, as will be elaborated throughout the text below. The most striking observation is that increasing $U$ pushes the onset scales $T_{orb}^{onset}$ and $T_{spin}^{onset}$ closer together until they essentially coincide. As a consequence, the Hund regime (small $U$) and the Mott regime (large $U$, close to the Mott transition), though adiabatically connected via a crossover regime, show dramatic differences for the temperature dependence of physical quantities (discussed below). The trends shown in the figure were extracted from our analysis of the 3HHM, but they match those found for $V_2O_3$ and $Sr_2RuO_4$ (see legend on the right), and we expect them to be generic for multi-orbital Mott and/or Hund systems.

In Mott systems, the Hubbard interaction is large, and at high temperatures the spectral function consists of a lower and a upper Hubbard band above and below the Fermi level, separated by a pseudogap, whose scale is set by the lowest atomic excitation energy, $E_{atomic}$. As the temperature is lowered, a resonance of resilient quasiparticles emerges inside the pseudogap (for a cartoon depiction, see Fig. 1, right inset), at a well-defined temperature scale, $T_M$, much below $E_{atomic}$, signaling the sudden appearance of mobile charge carriers. This behavior has been studied in detail in the one-band Hubbard model[32]. Here we propose that also for multi-orbital systems, the presence of a pseudogap at large temperatures, together with the abrupt emergence of a quasiparticle resonance at a temperature $T_M \ll E_{atomic}$, are fingerprints of Mott physics, distinguishing it from Hund physics. This Mott behavior is also reflected in the temperature dependence of various local susceptibilities. With decreasing temperature the static local charge susceptibility and local charge fluctuations first remain small and rather constant, while the local spin and orbital susceptibilities exhibit Curie behavior, indicative of unscreened local moments. Once the temperature drops below $T_M$, the appearance of mobile carriers causes the local charge susceptibility and charge fluctuations to increase, and the spin and orbital susceptibilities to deviate from pure Curie behavior, reflecting the onset of screening. For Mott systems, this onset thus occurs simultaneously for orbital and spin degrees of freedom, $T_{spin}^{onset} = T_{orb}^{onset} = T_M$.

The above signatures of Mott physics are in stark contrast to the behavior of Hund systems. These typically have much smaller values of $U$, and hence $E_{atomic}$. Consequently, the Hubbard side bands effectively overlap, so that the local density of states (LDOS) features a single incoherent peak even at temperatures as high as $E_{atomic}$ or beyond. This broad peak evolves into a coherent quasiparticle peak as the temperature is lowered (for a cartoon depiction, see Fig. 1, left inset). Due to the absence of a pseudogap at large temperatures, the local charge susceptibility is large already at high temperatures and increases continuously, but only slightly, with decreasing temperature. Strikingly, the local orbital

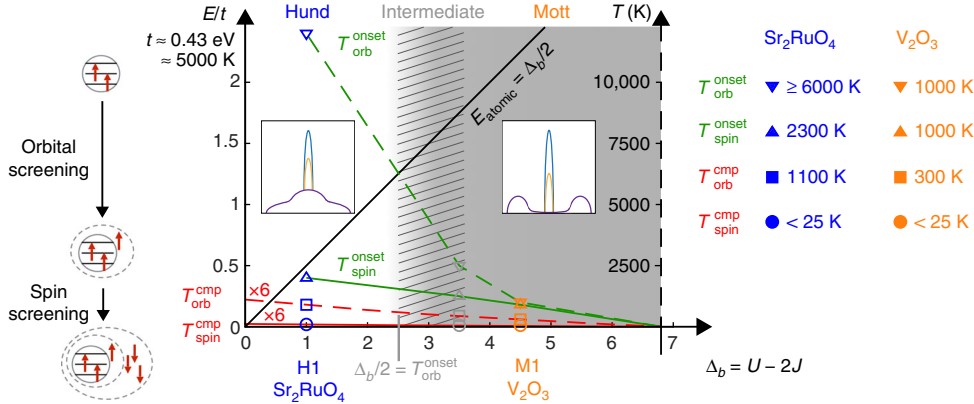

**Fig. 1** Schematic sketch of the behavior of four characteristic temperature scales. $T_{orb}^{onset}$ (green dashed), $T_{spin}^{onset}$ (green solid), $T_{orb}^{cmp}$ (red dashed), $T_{spin}^{cmp}$ (red solid) mark the onset and the completion of screening of orbital and spin degrees of freedom, respectively, as functions of the bare gap, $\Delta_b = U - 2J$, between the upper and lower Hubbard side band. White (gray) background indicates Hund (Mott) behavior at small (large) $\Delta_b$. A crossover between both (oblique gray lines) is found at intermediate $\Delta_b$. Open symbols on the left give the values of these scales as obtained from DMFT + NRG calculations for our three-band Hubbard–Hund model, with $J = 1$ and $U = 3$ (green), $U = 5.5$ (gray), and $U = 6.5$ (yellow). On the right, closed symbols give corresponding values obtained from DFT + DMFT calculations for the materials $Sr_2RuO_4$ (green) and $V_2O_3$ (yellow). Left: cartoon of two-stage screening for three local levels with $J \neq 0$, containing two electrons with total spin $S = 1$: with decreasing temperature, first orbital screening of the hole occurs, whereby a delocalized spin 1/2 combines with the local spin 1 to yield an orbital singlet with spin 3/2; then spin screening occurs, yielding an orbital and spin singlet[10]. The energy scales characterizing the two screening stages lie far apart for Hund systems, but close together for Mott systems. Insets: cartoons of the local density of states, $A(\omega)$, for a Hund system (left) and a Mott system close to the Mott transition (right), summarizing the essential differences in the evolution of the quasiparticle peak with decreasing temperature (purple to yellow to blue)

and spin susceptibilities show deviations from Curie-like behavior already at temperatures that are typically much higher than those in Mott systems. Moreover, orbital screening starts well before spin screening, $T_{orb}^{onset} \gg T_{spin}^{onset}$. Thus, Hund metals exhibit spin–orbital separation, featuring a broad temperature window, from $T_{orb}^{onset}$ down to $T_{spin}^{onset}$, involving screened, delocalized orbitals coupled to unscreened, localized spins. Importantly, $T_{orb}^{onset}$ can be much larger than $E_{atomic}$ in Hund metals, which is why no pseudogap appears even up to temperatures well above $E_{atomic}$ (it would appear only for $T \gtrsim T_{orb}^{onset}$, since it requires the breakdown of both spin and orbital screening). The fact that $T_{spin}^{onset}$ and $E_{atomic}$ are both $\ll T_{orb}^{onset}$ is a crucial difference relative to Mott systems. There, $T_{spin}^{onset} \simeq T_{orb}^{onset} \ll E_{atomic}$, so that the breakdown of spin and orbital screening, and the concomitant emergence of a pseudogap, is possible at temperatures well below $E_{atomic}$.

In principle, both multi-orbital Mott and Hund materials exhibit spin–orbital separation in the completion of screening: $\chi_{orb}$ crosses over to Pauli behavior at a larger temperature scale than $\chi_{spin}$, i.e. $T_{orb}^{cmp} \gg T_{spin}^{cmp}$. Since Fermi-liquid behavior occurs below $T_{spin}^{cmp}$, this scale can be identified with the Fermi-liquid scale $T_{FL} \equiv T_{spin}^{cmp}$. However, spin–orbital separation in the completion of screening is much more pronounced for the Hund material.

**Two archetypical materials $V_2O_3$ and $Sr_2RuO_4$.** We begin our discussion of the two example materials by summarizing some of their well-established properties. $V_2O_3$, a paramagnetic metal at ambient conditions, is proximate to an isostructural Mott transition (that can be induced by slightly Cr-doping), and a temperature-driven magnetic transition[15–17]. It exhibits Fermi-liquid behavior at low temperature when antiferromagnetism is quenched by doping or pressure[15–17]. $Sr_2RuO_4$, on the other hand, is a paramagnetic metal far away from a Mott insulating state[33]. As temperature decreases it shows Fermi-liquid behavior and eventually becomes superconducting at very low temperature[18]. Despite the very different distances to a Mott insulating

state, both materials have large specific heat coefficients in their Fermi-liquid states[15–18]. In both materials the observed Fermi-liquid scales are extremely low (around 25 K[15–17,34]), much smaller than the bare band energy or interaction parameters (order of eV). Pronounced quasiparticle peaks are observed in both materials using photoemission spectroscopy[35–38], and large values of mass renormalization are seen in $Sr_2RuO_4$ in various measurements[39–41]. Notably, the local physics on V/Ru sites are similar, with nominally two electrons/holes in three $t_{2g}$ orbitals. Due to the crystal field of the surrounding oxygen, the $t_{2g}$ orbitals of V are split into $e_g^\pi$ orbitals with two-fold degeneracy and an energetically higher-lying $a_{1g}$ orbital, while those of Ru are split into $xz/yz$ orbitals with two-fold degeneracy and an energetically lower-lying $xy$ orbital. Two electrons (holes) in three orbitals favor a spin-triplet $S = 1$ atomic state because of Hund's coupling in both $V_2O_3$[22,42–44] and $Sr_2RuO_4$[5].

We compute the spectra of the relevant correlated orbitals in $V_2O_3$ and $Sr_2RuO_4$ up to high temperature with DFT + DMFT. We have not taken into account the effects of the temperature-dependent changes in lattice parameters, which have been shown to be very important in materials near the Mott transition such as $V_2O_3$[45]. Nevertheless, the LDA + DMFT calculations here bring a degree of realism, such as band structure and crystal field effects, which is not present in the 3HHM calculations discussed further below. We focus first on the density of states at the Fermi level, estimated via $D(i\omega_0) = -\frac{1}{\pi} Im G(i\omega_0)$ ($\omega_0$ is the first Matsubara frequency, $G$ the computed local Green's function). Fig. 2a depicts the temperature dependence of $D(i\omega_0)$ for $e_g^\pi$ and $a_{1g}$ orbitals in $V_2O_3$. The results show that both orbitals share a characteristic temperature, $T_M = 1000$ K: $D(i\omega_0)$ is fairly flat at temperatures above $T_M$, which implies approximately rigid, i.e. temperature-independent, spectra. Below $T_M$, $D(i\omega_0)$ gradually acquires a larger magnitude in both orbitals as temperature is lowered, signaling the formation of a quasiparticle resonance. We note that it increases monotonically with decreasing temperature in the $e_g^\pi$ orbitals, but in the $a_{1g}$ orbital it first increases and then decreases a little. Thus at low temperature the density of states at the Fermi level has a dominant $e_g^\pi$ character. We emphasize that the

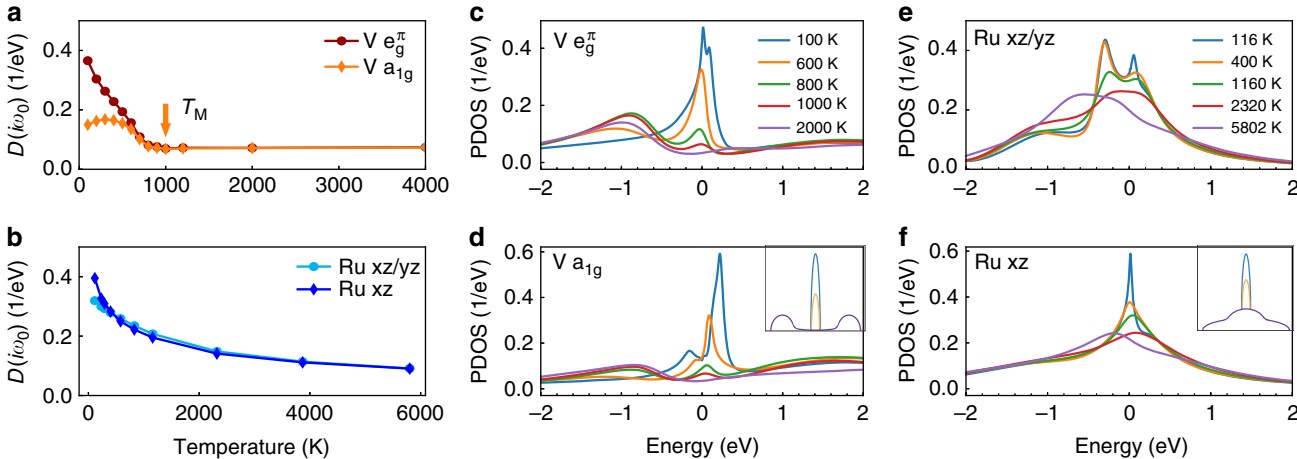

**Fig. 2** Local spectra of the correlated orbitals in $V_2O_3$ and $Sr_2RuO_4$. The results for $V_2O_3$ (**a**, **c**, **d**) and $Sr_2RuO_4$ (**b**, **e**, **f**) exhibit very different temperature-dependent behaviors. **a**, **b** The density of states at the Fermi level, estimated by $D(i\omega_0) = -\frac{1}{\pi}\text{Im}G(i\omega_0)$. **c**–**f** The correlated real-frequency spectra (PDOS), $A(\omega) = -\frac{1}{\pi}\text{Im}G(\omega)$. $D(i\omega_0)$ shows a suppression at a characteristic temperature $T_M = 1000$ K (indicated by the orange arrow) in $V_2O_3$ (**a**), while it evolves smoothly in $Sr_2RuO_4$ (**b**). As temperature decreases, in $V_2O_3$ the coherence resonance of both $e_g^\pi$ and $a_{1g}$ orbitals emerges from the pseudogap regime with low density of states between two incoherent peaks (**c**, **d**), while in $Sr_2RuO_4$ the coherence resonance of both $d_{xz/yz}$ and $d_{xy}$ orbitals emerges from a single broad incoherent peak with large finite density of states at the Fermi level (**e**, **f**). The insets in (**d**, **f**), repeated from Fig. 1, are cartoons of the temperature dependence of the Mott and Hund PDOS

evolution of $D(i\omega_0)$ is smooth and a first-order MIT is not involved. By contrast, in $Sr_2RuO_4$ the temperature dependence of the densities of states, $D(i\omega_0)$, of $d_{xz/yz}$ and $d_{xy}$ orbitals is very different, as depicted in Fig. 2b. For both orbitals, $D(i\omega_0)$ increases as temperature is decreased, with gradually increasing slope, showing no flat regime even at extremely high temperatures, where their values are already larger than those for $V_2O_3$ above $T_M$. In contrast to the case of $V_2O_3$, a quasiparticle resonance is present even at the highest temperatures studied and, thus, no characteristic temperature is found for its onset, as discussed in the next paragraph.

We also study the correlated real-frequency projected density of states (PDOS), $A(\omega) = -\frac{1}{\pi}\text{Im}G(\omega)$, for the different orbitals. These are obtained by analytically continuing the computed Matsubara self-energy and then computing the local Green's function. The results for $V_2O_3$ are depicted in Fig. 2c, d. At very high temperatures, we observe a typical Mott feature: a pseudogap exists at the Fermi level, between two broad humps in the incoherent spectra, with maxima near −1 and 2 eV. With decreasing temperature spectral weight is transferred from the high-energy humps into the pseudogap and a quasiparticle peak emerges similarly in both orbitals at the Fermi level on top of the pseudogap. The characteristic temperature for the onset of the formation of the coherence resonance is roughly consistent with $T_M = 1000$ K determined above. As temperature decreases further, the magnitude of the coherence peak in both orbitals increases gradually, and at very low temperature both orbitals show a coherence resonance with a pronounced, thin cusp. In the $e_g^\pi$ orbitals the resonance is peaked at the Fermi level while the $a_{1g}$ quasiparticle peak slightly moves away from the Fermi level when the temperature is lowered, thus reducing the density of states at the Fermi level. The resulting temperature evolution of the zero-frequency density of states in both orbitals is consistent with the $D(i\omega_0)$ discussed above, including the non-monotonic behavior of the $a_{1g}$ orbital in Fig. 2a. For $Sr_2RuO_4$ the slow increase of the density of states at the Fermi level, $D(i\omega_0)$, with decreasing temperature becomes clear from the PDOS, shown in Fig. 2e, f. The correlated high-temperature local spectra are characterized by a single broad feature (no side-humps), which shifts its

position slightly towards the Fermi level with decreasing temperature, while its shape remains almost unchanged. This is very different from the spectra in $V_2O_3$, which at high temperatures show a pseudogap between two broad side peaks. When the temperature is decreased further, a sharp narrow peak gradually develops in both orbitals from the broad, incoherent feature. In this process only a small fraction of spectral weight is transferred from higher frequencies to a 1 eV range around the Fermi level. At low temperature, the spectra of both $d_{xz/yz}$ and $d_{xy}$ orbitals are similar to their corresponding DFT values with a renormalized bandwidth and show a pronounced, thin cusp as in the case of $V_2O_3$.

The different temperature dependences of the local spectra of $V_2O_3$ and $Sr_2RuO_4$ can be viewed as fingerprints distinguishing Mott from Hund systems, respectively. With decreasing temperature the quasiparticle resonance of $V_2O_3$ emerges from a high-temperature pseudogap regime with very low density of states between incoherent spectra (see purple curve in the cartoon in inset of Fig. 2d). This is consistent with the widely held belief that Mott physics governs $V_2O_3$. It is described by a single characteristic temperature scale, $T_M$, which indicates the onset of formation of the quasiparticle resonance. By contrast, for $Sr_2RuO_4$ the quasiparticle resonance develops with decreasing temperature from a single incoherent peak that has a large value at the Fermi level already at very high temperature (see purple curve in the cartoon in inset of Fig. 2f). The demonstration of these two distinct routes towards forming the coherent Fermi-liquid at low temperature is one of the main results of this work.

We next consider the static local charge susceptibility, $\chi_{\text{charge}} = \int_0^\beta \langle N_d(\tau)N_d(0)\rangle d\tau - \beta\langle N_d\rangle^2$, and the local charge fluctuations, $\langle \Delta N^2 \rangle = \langle N_d^2 \rangle - \langle N_d \rangle^2$, shown in Fig. 3a, b, respectively. ($N_d$ is the total occupancy of $t_{2g}$ orbitals.) For both materials, the behavior of $\chi_{\text{charge}}$ mimics that of $\langle \Delta N^2 \rangle$, hence we focus on the latter below. $\langle \Delta N^2 \rangle$ is much smaller, with a much stronger temperature dependence, in $V_2O_3$ than in $Sr_2RuO_4$. For $V_2O_3$, $\langle \Delta N^2 \rangle$ initially remains small and almost constant with decreasing temperature, signifying the suppression of charge fluctuations in the pseudogap regime. It then increases rather abruptly, signifying the onset of charge delocalization, at the same

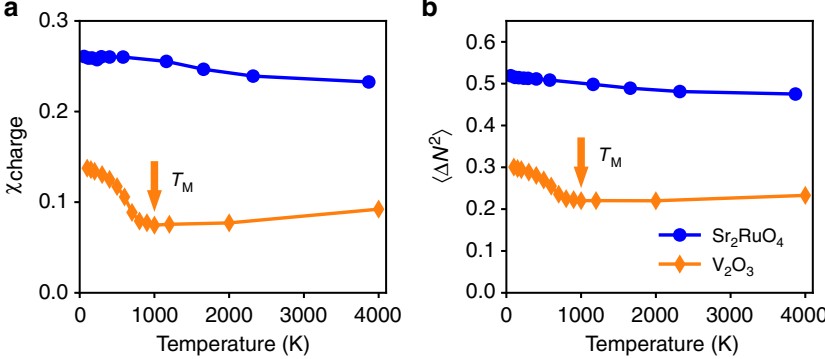

**Fig. 3** The static local charge susceptibility $\chi_{charge}$, and local charge fluctuation $\langle \Delta N^2 \rangle$. (**a**) $\chi_{charge}$ and (**b**) $\langle \Delta N^2 \rangle$ is computed for $V_2O_3$ (orange diamonds) and $Sr_2RuO_4$ (blue circles). In the Hund system $Sr_2RuO_4$ both $\chi_{charge}$ and $\langle \Delta N^2 \rangle$ are large and only weakly dependent on temperature. By contrast, in the Mott system $V_2O_3$ they are much smaller and strongly temperature dependent. The orange arrows indicate that in $V_2O_3$ the minima of the local charge susceptibility and fluctuation occur at the same temperature scale, $T_M$, as that marking the emergence of the quasiparticle peak in the local PDOS

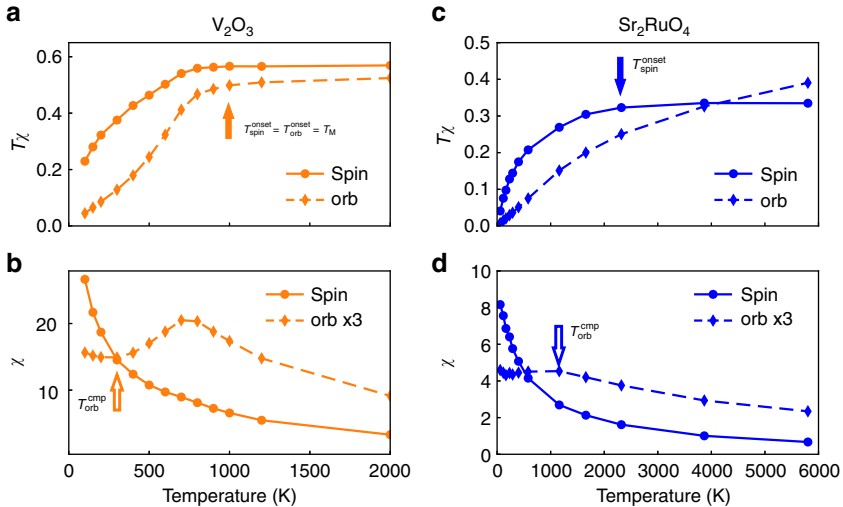

**Fig. 4** The static local orbital and spin susceptibilities $\chi_{orb}$ and $\chi_{spin}$ plotted as functions of temperature. $T\chi$ and $\chi$ are shown in the upper and lower panels, respectively, for $V_2O_3$ (**a, b**) and $Sr_2RuO_4$ (**c, d**). The Curie law holds above 1000 K (indicated by filled orange arrow) in the spin and orbital susceptibility of $V_2O_3$ (**a**). In $Sr_2RuO_4$ the spin susceptibility follows Curie-like behavior above 2300 K (indicated by filled blue arrow), while the orbital susceptibility does not follow a Curie law in the temperature range studied (**c**). The spin susceptibility of both materials (**b, d**) does not saturate at the lowest accessible temperature, indicating an even lower Fermi-liquid scale. The orbital susceptibility has only weak temperature dependence below 300 K in $V_2O_3$ (**b**) and below 1100 K in $Sr_2RuO_4$ (**d**) (indicated by open orange and blue arrows, respectively)

temperature, $T_M = 1000$ K (orange arrow in Fig. 3a), as that where the quasiparticle peak begins to emerge. By contrast, for $Sr_2RuO_4$ $\langle \Delta N^2 \rangle$ exhibits only a weak temperature dependence, persisting up to the highest temperature studied but changing by <10% over this range. We have also computed the static local spin and orbital susceptibilities, defined as $\chi_{spin} = \int_0^\beta \langle S_z(\tau) S_z(0) \rangle d\tau$ and $\chi_{orb} = \int_0^\beta \langle \Delta N_{orb}(\tau) \Delta N_{orb} \rangle - \beta \langle \Delta N_{orb} \rangle^2$, where $S_z$ is the total spin momentum in the $t_{2g}$ orbitals, $\Delta N_{orb} = N_a/2 - N_b$ is the occupancy difference per orbital, and $(a, b)$ denotes $(e_g^\pi, a_{1g})$ in $V_2O_3$ and $(xz/yz, xy)$ in $Sr_2RuO_4$, respectively. The results are depicted in Fig. 4. In $V_2O_3$, both the spin and orbital susceptibilities exhibit Curie behavior, i.e. $T\chi_{spin}$ and $T\chi_{orb}$ are approximately constant at high temperature (Fig. 4a). Notably, with decreasing temperature deviations from the Curie behavior set in at the same characteristic temperature, $T_M = 1000$ K, as that determined above from the local PDOS evolution. Thus, spin and orbital degrees of freedom start to be screened simultaneously with the formation of a coherence resonance in the prototype

Mott system $V_2O_3$, $T_{orb}^{onset} = T_{spin}^{onset} = T_M$. By contrast, in the Hund material $Sr_2RuO_4$, Curie-like behavior in the spin susceptibility is seen only at very high temperatures. With decreasing temperature, it ceases already at around $T_{spin}^{onset} \simeq 2300$ K (Fig. 4c), a scale much higher than that in $V_2O_3$. For the orbital susceptibility the situation is even more extreme: it does not show Curie behavior even at the highest temperature studied $(T_{orb}^{onset} \geq 6000$ K$)$. This is evidence of spin–orbital separation in $Sr_2RuO_4$: the screening of the orbital degrees of freedom starts at much higher temperature than that of the spin degrees of freedom, $T_{orb}^{onset} \gg T_{spin}^{onset}$. Hence the onset of deviations from Curie-like behavior in the spin/orbital susceptibility, i.e. the onset of screening of spin/orbital degrees of freedom, is very different in $V_2O_3$ and $Sr_2RuO_4$. These differences constitute another set of fingerprints distinguishing Mott from Hund systems. It will be further analyzed below in the context of our 3HHM calculations.

Next we discuss the completion of orbital and spin screening, characterized by the temperature scales, $T_{orb}^{cmp}$ and $T_{spin}^{cmp}$, below

which the corresponding susceptibilities become constant. In both materials, $\chi_{\text{orb}}$ seems to become essentially constant at low temperatures, with the orbital screening completion scale in $V_2O_3$, $T_{\text{orb}}^{\text{cmp}} \simeq 300$ K (Fig. 4b), being much smaller than in $Sr_2RuO_4$, $T_{\text{orb}}^{\text{cmp}} \simeq 1100$ K (Fig. 4d). By contrast, in both materials the spin susceptibility increases with decreasing temperature, and is not fully screened even at the lowest temperature studied. This is consistent with the experimental observations that in both materials $T_{\text{FL}}$ is as low as about 25 K, and $T_{\text{FL}}$ provides an estimation for $T_{\text{spin}}^{\text{cmp}}$ at which the spin degrees of freedom are fully screened. In summary, we clearly deduce spin–orbital separation in the completion of screening, $T_{\text{orb}}^{\text{cmp}} \gg T_{\text{spin}}^{\text{cmp}}$, for the Hund metal $Sr_2RuO_4$, while this effect is less pronounced in the Mott material $V_2O_3$, where $\chi_{\text{orb}}$ in addition shows a bumb before it tends to saturate at lower temperatures.

In Hund metals the spin–orbital separation has been pointed out in numerical studies of the frequency dependence of the local self-energy and susceptibilities[9–11] and in an analytical estimate of the Kondo scales[13]. Here, our results reveal that it also occurs in the temperature domain. We note that our computed spin susceptibility of $Sr_2RuO_4$ is consistent with earlier results using a narrower temperature range[14].

In both materials the entropy of the correlated atom reaches a plateau of $\ln(3 \times 3 = 9)$ at high temperatures as expected for a high spin ($S = 1$) state with large contribution from three active $t_{2g}$ orbital degrees of freedom[8,9]. Notably, in Mott systems, with decreasing temperature the plateau persists down to the temperature scale, $T_M$, until which both the spin and orbital degrees of freedom remain unquenched and the quasiparticle resonance has not yet formed in the pseudogap of the local correlated spectrum. These results are presented in the Supplementary Fig. 1.

### Three-band Hubbard–Hund model.

We now turn to the 3HHM, described by the Hamiltonian

$$\hat{H} = \sum_i \left( -\mu \hat{N}_i + \hat{H}_{\text{int}}[\hat{d}_{i\nu}^\dagger] \right) + \sum_{\langle ij \rangle \nu} t\, \hat{d}_{i\nu}^\dagger \hat{d}_{j\nu},$$

$$\hat{H}_{\text{int}}[\hat{d}_{i\nu}^\dagger] = \tfrac{1}{2}\left(U - \tfrac{3}{2}J\right)\hat{N}_i(\hat{N}_i - 1) - J\hat{\mathbf{S}}_i^2 + \tfrac{3}{4}J\hat{N}_i. \quad (1)$$

The on-site interaction term incorporates Mott and Hund physics through $U$ and $J$, respectively. $\hat{d}_{i\nu}^\dagger$ creates an electron on site $i$ of flavor $\nu = (m\sigma)$, which is composed of a spin ($\sigma = \uparrow, \downarrow$) and orbital ($m = 1, 2, 3$) index. $\hat{n}_{i\nu} = \hat{d}_{i\nu}^\dagger \hat{d}_{i\nu}$ counts the electrons of flavor $\nu$ on site $i$. $\hat{N}_i = \sum_\nu \hat{n}_{i\nu}$ is the total number operator for site $i$ and $\hat{\mathbf{S}}_i$ its total spin, with components $\hat{S}_i^\alpha = \sum_{m\sigma\sigma'} \hat{d}_{im\sigma}^\dagger \tfrac{1}{2}\sigma_{\sigma\sigma'}^\alpha \hat{d}_{im\sigma'}$, where $\sigma^\alpha$ are Pauli matrices. We take a uniform hopping amplitude, $t = 1$, serving as energy unit in the 3HHM, and a Bethe lattice in the limit of large lattice coordination. The total width of each of the degenerate bands is $W = 4$. We choose the chemical potential $\mu$ such that the total filling per lattice site is $\langle N_i \rangle = 2$. The model is solved numerically exactly using DMFT + NRG[9,10].

The 3HHM enables the exploration of a broad region of parameters at arbitrary low temperatures. Fig. 5a illustrates the $J$–$U$ phase diagram at $T = 0$. To illustrate the difference between large and small $U$, and non-zero and zero $J$, we will focus on four parameter combinations, denoted by M1, H1, M0, and W0, depicted by asterisks in Fig. 5a, and defined in detail in the figure caption. The Mott system M1 ($U = 6.5$) and the Hund system H1 ($U = 3$), both with $J = 1$, lie close to or far from the Mott transition and qualitatively mimic $V_2O_3$ and $Sr_2RuO_4$, respectively, considering their multi-orbital nature, sizable Hund's coupling and distances to the Mott transition. The Mott system

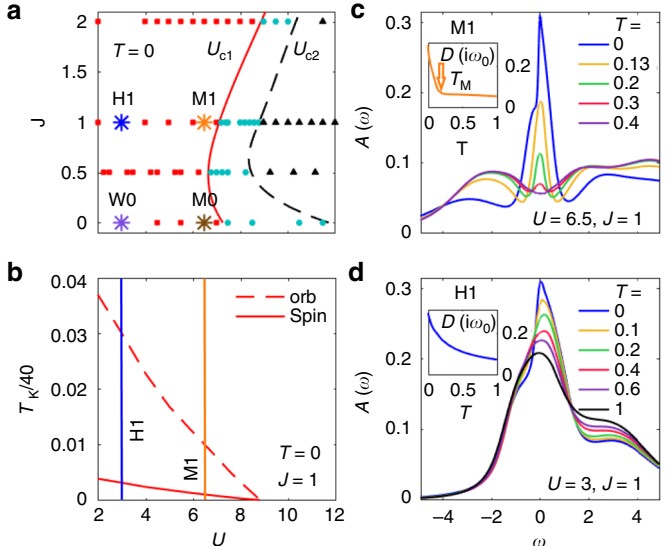

**Fig. 5** Disentangling features of Mott and Hund physics in a DMFT + NRG study of the 3HHM: (**a**) phase diagram, (**b**) Kondo temperatures, and (**c, d**) LDOS $A(\omega)$. **a** The $T = 0$ phase diagram (cf. Fig. 5 of ref. [10]) reveals three phases in the $J$-$U$-plane: a metallic phase (red squares), a coexistence region (blue circles), and an insulating phase (black triangles), separated by two-phase transition lines $U_{c1}$ (solid red curve), and $U_{c2}$ (dashed black curve), respectively. In panels (**c, d**) and also in Fig. 6 we focus on four parameter combinations, indicated in (**a**) by colored asterisks: two Mott systems with $U = 6.5$ near the $U_{c1}$ phase transition line, with $J = 1$ (M1) or $J = 0$ (M0); and two systems with $U = 3$ far from the transition and deep in the metallic state, a Hund system with $J = 1$ (H1) and a weakly correlated system with $J = 0$ (W0). **b** The Kondo temperatures, here shown for $J = 1$ (cf. Fig. 12 of ref. [10]) are extracted from frequency-dependent susceptibilities at $T = 0$, as defined in ref. [9]. $T_K^{\text{spin}}$ ($T_K^{\text{orb}}$) corresponds to the screening of spin (orbital) degrees of freedom. Fermi liquid behavior sets in below the temperature scale $T_K^{\text{spin}}/40 \approx T_{\text{spin}}^{\text{cmp}}$. Orange and blue vertical lines mark the values of $U$ used for M1 and H1, respectively. **c, d** The temperature dependence of the LDOS for M1 and H1, respectively. The energy scale of the lowest bare atomic excitations, $\pm E_{\text{atomic}} = \pm \left(\tfrac{1}{2}U - J\right)$, and thus the Hubbard bands, is much larger for M1 than H1. For M1 in (**c**) a pseudogap (a typical Mott feature) emerges when the temperature increases past a characteristic value, $T_M$, which lies far below the rather large scale $E_{\text{atomic}}^{\text{M1}} \simeq 2.25$. By contrast, for H1 in (**d**) a pronounced peak in the density of states still exists even at very high temperatures, $T > 0.5$, that exceed the rather small scale $E_{\text{atomic}}^{\text{H1}} \simeq 0.5$. The insets of (**c, d**) show the LDOS at the Fermi level, estimated by $D(i\omega_0) = -\tfrac{1}{\pi}\text{Im}G(i\omega_0)$, for M1 (orange) and H1 (blue)

M0 ($U = 6.5$) and the weakly correlated system W0 ($U = 3$), both with $J = 0$, illustrate the consequences of turning off Hund's coupling altogether.

Figure 5 displays the LDOS, $A(\omega) = -(1/\pi)\text{Im}G(\omega)$, for M1 and H1 (Fig. 5c, d), and the corresponding density of states at the Fermi level (insets of Fig. 5c, d), estimated by $D(i\omega_0) = -\tfrac{1}{\pi}\text{Im}G(i\omega_0)$. Figure 6 shows the static local susceptibilities $T\chi$ (a, c) and $\chi \equiv \chi_d(\omega = 0)$ (Fig. 6b, d) for the spin (solid) and orbital (dashed) degrees of freedom of M1 (orange), M0 (brown), H1 (blue), and H0 (purple). The corresponding dynamical real-frequency spin and orbital susceptibilities are defined as $\chi_{d,\text{spin}}(\omega) = \tfrac{1}{3}\sum_\alpha \langle \hat{S}^\alpha || \hat{S}^\alpha \rangle_\omega$ and $\chi_{d,\text{orb}}(\omega) = \tfrac{1}{8}\sum_a \langle \hat{T}^a || \hat{T}^a \rangle_\omega$, respectively[46,47], where $\hat{T}^a = \sum_{mm'\sigma} \hat{d}_{m\sigma}^\dagger \tfrac{1}{2}\tau_{mm'}^a \hat{d}_{m'\sigma}$ are the impurity orbital operators with the SU(3) Gell–Mann matrices, $\tau^a$, normalized as $\text{Tr}[\tau^a\tau^b] = 2\delta_{ab}$. The qualitative similarities of Figs. 5c, d and 6 with those

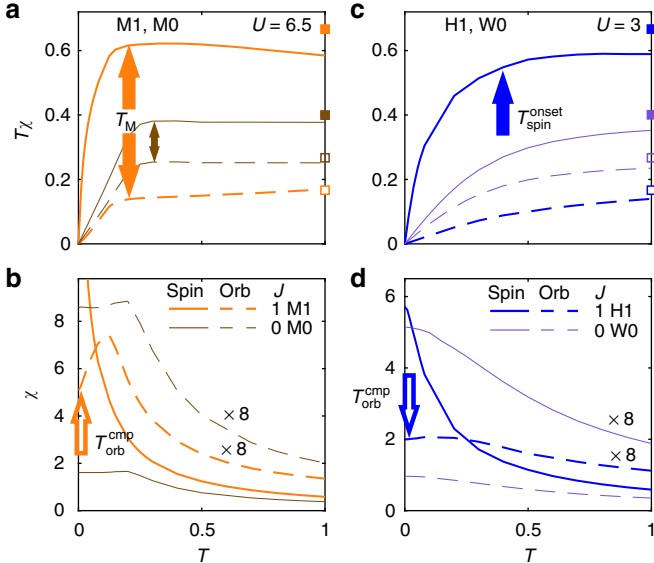

**Fig. 6** Disentangling features of Mott and Hund physics in a DMFT + NRG study of the 3HHM: static local susceptibilities. The local spin and orbital susceptibilities are shown as functions of $T$ for M1 (orange) and M0 (brown) in (**a**, **b**), and for H1 (blue) and W0 (purple) in (**c**, **d**), with $T\chi$ depicted in the upper panels (**a**, **c**) and $\chi$ in the lower panels (**b**, **d**). **a**, **c** For temperatures well above $T_{\rm orb}^{\rm onset}$ or $T_{\rm spin}^{\rm onset}$, respectively, $T\chi_{\rm orb}$ and $T\chi_{\rm spin}$ approach plateaus, indicative of a Curie law, as expected for unscreened spin or orbital degrees of freedom. The observed plateau heights are roughly comparable to the values expected[46] for free local moments with occupancy strictly equal to 2 and, for M1 and H1 (M0 and W0), spin equal to 1 (and 0), for which $T\chi_{\rm spin}^{\rm free} = \frac{1}{3}\langle\hat{S}^2\rangle = 2/3\,(2/5)$ and $T\chi_{\rm orb}^{\rm free} = \frac{1}{8}\langle\hat{T}^2\rangle = 1/6\,(4/15)$ indicated by filled and empty squares on the right vertical axes, respectively. (Deviations of the observed plateaus from these local moment values reflect admixtures of states with different occupancy or spin, see Supplementary Fig. 3.) For M1 in (**a**), the Curie law ceases to hold for both $\chi_{\rm orb}$ and $\chi_{\rm spin}$ below about $T_{\rm M} \simeq 0.2$ (filled orange arrows). For H1 in (**c**), $\chi_{\rm spin}$ deviates from a Curie-like behavior below about $T \simeq 0.4$ (filled blue arrow), while $\chi_{\rm orb}$ does not follow a Curie law in the temperature range displayed. For M0 in (**a**) and W0 in (**c**), deviations from Curie behavior set in at similar temperatures for $\chi_{\rm spin}$ and $\chi_{\rm orb}$, since $\chi_{\rm spin} = 1.5\chi_{\rm orb}$ for $J = 0$ (M0: small brown double arrow, W0: outside temperatures range of plot). Thus the onset of screening shows spin-orbital separation for H1, but not for M1 (due to its proximity to the Mott transition), and also not for M0 and W0 (since these have $J = 0$). **b**, **d** For both M1 and H1, $\chi_{\rm spin}$ saturates at very low Fermi-liquid temperatures (not displayed here, but clearly deducible from the underlying zero-temperature NRG data[9,10]). By contrast, $\chi_{\rm orb}$ is approximately temperature independent below $T = 0.01$ (open orange arrow) for M1 in (**b**) and below $T = 0.03$ (open blue arrow) for H1 in (**d**). For M0 in (**b**) and W0 in (**d**), $\chi_{\rm orb}$ and $\chi_{\rm spin}$ become temperature independent at similar temperatures. Thus, the completion of screening shows tendencies of spin-orbital separation for M1 and H1 (since $J \neq 0$), but not for M0 and W0 (since $J = 0$). Moreover, $T_{\rm spin}^{\rm cmp}$ and hence $T_{\rm FL}$ is much smaller for $J \neq 0$ than for $J = 0$

in Figs. 2 and 4, respectively, are obvious, in spite of the simplified band structure and the absence of crystal fields. Let us now discuss these in detail.

**Mott system M1.** The basic features of V$_2$O$_3$ are reproduced by M1, lying close to the phase transition line. At high temperatures we also observe a pseudogap in the incoherent spectra at the Fermi level, formed between two broad Hubbard sidebands, one at negative and one with minor substructure at positive frequencies, respectively (Fig. 5c, red and purple curves). With

decreasing temperature spectral weight is transferred from these high-energy humps into the pseudogap, building up a clear peak at about $T_{\rm M} \simeq 0.2$, which evolves into a pronounced, sharp coherence resonance at very low temperature (Fig. 5c, blue curve). This behavior is confirmed by $D(i\omega_0)$ (inset of Fig. 5c). $T\chi$ shows flat Curie behavior for both orbital and spin degrees of freedom in the pseudogapped phase at high temperatures. With decreasing temperature the orbitals and spins start to get screened simultaneously at the same energy scale, $T_{\rm orb}^{\rm onset} = T_{\rm spin}^{\rm onset} = T_{\rm M}$ (Fig. 6a, orange arrow), at which the resonance emerges in the pseudogap, analogously to the behavior in the Mott material V$_2$O$_3$.

**Hund system H1.** For the Hund system H1 far from the phase transition line, the physical properties (LDOS, local spin and orbital susceptibility) show the same qualitative behavior as for Sr$_2$RuO$_4$. At very high temperatures (above $T_{\rm spin}^{\rm onset} \simeq 0.4$, red, purple and black curves in Fig. 5d) the local spectral function has a large density of states near the Fermi energy, in contrast to the pseudogap present for M1. At these large temperatures the spin susceptibility shows Curie-like behavior, in that $T\chi_{\rm spin}$ is essentially constant there, whereas $T\chi_{\rm orb}$ decreases with decreasing temperature (Fig. 6c). This indicates that the spins are still large and (quasi-)free while the orbitals are already being screened, $T_{\rm spin}^{\rm onset} \ll T_{\rm orb}^{\rm onset}$. Below $T_{\rm spin}^{\rm onset}$ also the spin degrees of freedom get screened and a pronounced quasiparticle peak gradually develops, with a sharp cusp at low frequencies and very low temperatures (blue curve in Fig. 5d). This spin–orbital separation of onset scales is a decisive fingerprint of Hund systems: it is absent for M1 and, for $J = 0$, it is absent for both W0 and M0, i.e. independently of the degree of correlations. In contrast to M1, $D(i\omega_0)$ for H1 is large already at high temperatures and increases continuously with decreasing temperature (inset of Fig. 5d).

**Completion of screening.** In principle, for both M1 and H1, i.e. both close to and far from the Mott transition, orbital screening is completed at a higher temperature than spin screening. Indeed, an approximately temperature-independent, Pauli-like susceptibility, is observed for $\chi_{\rm orb}$ (dashed lines) below $T_{\rm orb}^{\rm cmp}$ (indicated by open arrows in Fig. 6b, d), while $\chi_{\rm spin}$ (solid lines) still increase with decreasing temperature, hence $T_{\rm orb}^{\rm cmp} \gg T_{\rm spin}^{\rm cmp}$. This effect is more pronounced for H1, i.e. far from the Mott transition. By contrast, the corresponding M0 and W0 curves in Fig. 6b, d, having $J = 0$, show no spin–orbital separation for the completion of screening, i.e. $T_{\rm orb}^{\rm cmp} \simeq T_{\rm spin}^{\rm cmp}$, as described in more detail in the figure caption.

**Discussion**
The DMFT solution of the 3HHM with nonzero $J$ enables us to understand the interplay between Mott and Hund physics and its materials manifestations from an impurity model perspective. Far from the transition, a picture in terms of a multi-orbital Kondo model in a broad-bandwidth metallic bath applies. Standard analysis of the logarithmic Kondo singularies showed that $T_{\rm spin}^{\rm onset} \ll T_{\rm orb}^{\rm onset}$[13]. As we approach the Mott boundary, charge fluctuations are blocked, resulting in well-separated Hubbard bands. Here the onset of the Kondo resonance is not signaled by logarithmic singularities but instead it is driven by the DMFT-self-consistency condition[48]. In this regime the onsets of screening for spin and orbital degrees of freedom occur at the same scale, namely that where charge delocalization sets in. The spin–orbital separation in the completion of screening, which occurs at low temperatures in both Mott and Hund systems, can be understood from a zero-temperature analysis of the 3HHM. We define characteristic Kondo scales $T_{\rm K}^{\rm orb}$ and $T_{\rm K}^{\rm spin}$, from the

maximum in the zero-temperature, frequency-dependent local orbital and spin susceptibilities[9], respectively, and display them in Fig. 5b for $J = 1$ as a function of $U$. We find that $T_K^{spin} \ll T_K^{orb}$, so an intermediate region with free spins and quenched orbitals is a generic feature of multi-orbital systems with significant Hund's coupling as was surmised from earlier studies. For both the Hund and Mott system results we deduce that $T_K/40 \approx T^{cmp}$: spin–orbital separation in frequency space thus has a direct manifestation in the completion of screening as a function of temperature. Below the spin completion scale we have a Fermi liquid. As we approach the Mott boundary for increasing $U$ the spin–orbital separation region shrinks, and all the energy scales are reduced, as shown in Fig. 5b, elucidating the reduced $T^{cmp}_{orb}$ in the Mott system (and in $V_2O_3$) compared to the Hund system (and $Sr_2RuO_4$).

Finally, we note that the systems M0 and W0, with $J = 0$, show no spin–orbital separation for the onset or completion of screening (see Fig. 6a–d and their discussion in the figure caption). Conversely, turning on $J$ pushes the Fermi liquid scale $T_{FL} = T^{cmp}_{spin}$ strongly downward relative to $T^{onset}_{orb}$. This significantly reduces the quasiparticle weight $Z = m/m^*$ (which is proportional to $T_{FL}$[10]) and increases the strength of correlations. Hence the Hund system H1 is much more strongly correlated than W0, although $U$ is the same for both. These differences leave clear fingerprints in photoemission spectra, where $Z$ characterizes the slope of the quasiparticle dispersion. Moreover, the shape of the quasiparticle peak shows substructure indicative of spin–orbital separation for sizable $J$, but not for $J = 0$. For a detailed illustration of these points, see Supplementary Fig. 2.

In conclusion, we revealed contrasting signatures of Mottness and Hundness in two archetypal materials, $V_2O_3$ and $Sr_2RuO_4$, in the formation of the quasiparticle resonance in the local correlated spectra, and in the temperature dependence of the charge, spin, and orbital susceptibility as well as the impurity entropy. Mott and Hund physics manifest in the process in which the atomic degrees of freedom at high energies evolve towards low energies to form fermionic quasiparticles. We highlight the observation of four temperature scales that characterize the onset and the completion of screening of the spin and the orbital degrees of freedom. We find that a non-zero Hund's coupling leads to spin–orbital separation in the completion of screening at low temperatures, $T^{cmp}_{orb} \gg T^{cmp}_{spin}$, and this is more pronounced for Hund systems. However, Mott and Hund systems show contrasting behavior at intermediate to high energies, due to the very different relations of their overall quasiparticle peak width and their atomic excitation scales: $T^{onset}_{orb} \ll E_{atomic}$ for Mott systems vs. $E_{atomic} \lesssim T^{onset}_{orb}$ for Hund systems. In the Mott system $V_2O_3$ the strong Coulomb repulsion localizes the charge at high temperature, with decreasing temperature the onset of charge localization triggers the simultaneous onset of the screening of the spin and orbital degrees, accompanied by the formation of the coherence resonance at $T_M \equiv T^{onset}_{spin} = T^{onset}_{orb} \ll E_{atomic}$. In contrast, in $Sr_2RuO_4$ Coulomb repulsion is much weaker, so that no charge localization occurs even at very high temperatures. Therefore, charge fluctuations triggering the onset of screening are possible even at high temperatures, leading—due to the presence of sizeable Hund's coupling—to a clear separation in the energy scales at which this screening sets in for spin and orbital fluctuations, with $T^{onset}_{spin} \ll T^{onset}_{orb}$. All these findings are generic and do not depend on microscopic details. They only require a sizeable Hund's coupling, and are controlled by the distance to the Mott localization boundary. This is confirmed by a DMFT + NRG study of a model three-band Hubbard–Hund Hamiltonian, thus establishing a general phenomenology of Mottness and Hundness in multi-orbital systems. Our results give not only new perspectives

into the archetypical strongly correlated materials, $V_2O_3$ and $Sr_2RuO_4$, but will be useful in interpreting experimental measurements on other correlated metals and in identifying the origin of their correlations.

## Methods

**DFT + DMFT + CTQMC.** The two prototype materials are investigated using the all-electron DMFT method as implemented in ref. [49] based on the WIEN2k package[50] and the continuous-time quantum Monte-Carlo (CTQMC) impurity solver[51,52]. We used projectors within a large (20 eV) energy window, i.e. we used a high-energy cutoff scale, to construct local orbitals, thus the oxygen orbitals hybridizing with the $d$ orbitals were explicitly included. With such a large energy window the resulting $d$ orbitals are very localized. In our two example materials these are the $t_{2g}$ levels of Ru and V atoms, which we treated dynamically with DMFT, all other states were treated statically and no states were eliminated in the calculations. The nominal double counting scheme with the form $\Sigma_{DC} = U(n_{imp} - 1/2) - \frac{1}{2}J(n_{imp} - 1)$ was used, where $n_{imp}$ is the nominal occupancy of $d$ orbitals. The onsite interactions in terms of Coulomb interaction $U$ and Hund's coupling $J$ were chosen to be $(U, J) = (6.0, 0.8)$ eV for V in $V_2O_3$ and $(U, J) = (4.5, 1.0)$ eV for Ru in $Sr_2RuO_4$. The impurity entropy was computed by integrating the impurity internal energy up to high temperature, following ref. [53]. Our DFT + DMFT setup was successful in describing the correlation effects in both materials[28,31]. It captures the phase diagram of $V_2O_3$ which exhibits a Mott MIT and our computed electronic structure is consistent with experimental measurements[28]. The approach also describes the electronic structure of $Sr_2RuO_4$ and is in good agreement with the results of experimental measurements[31] and other DFT + DMFT calculations[5,29,30]. In addition, our studies[28,31] correctly characterize the transport and optical properties of both materials. These successes gave us confidence to extend our studies to even higher temperatures, and for quantities which have yet to be measured experimentally.

**DMFT + NRG.** We solved the 3HHM using DMFT[19] in combination with an efficient multi-band impurity solver[9,10], the full-density-matrix (fdm) NRG[54]. Our fdmNRG solver employs a complete basis set[55,56], constructed from the discarded states of all NRG iterations. Spectral functions for the discretized model are given from the Lehmann representation as a sum of poles, and can be calculated accurately directly on the real axis in sum-rule conserving fashion[57] at zero or arbitrary finite temperature. Continuous spectra are obtained by broadening the discrete data with a standard log-gaussian Kernel of frequency-dependent[54,58] width. Further, fdmNRG is implemented in the unified tensor representation of the QSpace approach[47] that allows us to exploit Abelian and non-Abelian symmetries on a generic level (here $U(1)_{charge} \times SU(2)_{spin} \times SU(3)_{orb}$). For further details of our DMFT + NRG calculations see the Supplementary material of ref. [9].

## Data availability

The authors declare that the data supporting the findings of this study are available within the paper (and its Supplementary Information).

## Code availability

Computer codes and algorithms to generate the results of this article are available upon request.

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

## Acknowledgements

Work by X.D. and G.K. was supported by NSF DMR-1733071. Work by K.H. was supported by NSF DMR 1405303. K.M.S. and j.v.D. acknowledge support from the excellence initiative NIM; A. W. was supported by the U.S. Department of Energy, Office of Basic Energy Sciences, under Contract No. DE-SC0012704.

## Author contributions

X.D., K.M.S., and G.K. proposed this project; X.D. performed the DFT + DMFT calculations and analyzed the results together with G.K. and K.H. K.H. developed the DFT + DMFT code used and and assisted the computation setup; K.M.S. performed the DMFT + NRG calculations; A.W. developed the NRG code and assisted K.M.S. in the initial stages of the DMFT + NRG computation. X.D. and K.M.S. drafted the manuscript with the help of G.K., K.H., A.W., and J.v.D.

**Additional information**

**Competing interests:** The authors declare no competing interests.

