## [Peer Review File · Nature Communications]

Reviewers' comments:

Reviewer #1 (Remarks to the Author):

Referee report for

"Signature of Mottness and Hundness archetypal correlated metals"

by Deng et al.

Submitted for consideration as an article in Nature Communication.

This article relates DFT+DMFT simulations of two materials, V₂O₃ and Sr₂RuO₄, which are taken as archetypes for Mott and Hund's correlated electronic physics in metals.

The article bases the simulations on parameters (mainly interaction couplings U and J) estimated in previous calculations from the same authors Ref. 23 and 26. Presented results focus on the temperature dependence of local spectral functions, charge, spin and orbital local susceptibilities, and impurity entropy. The authors highlight crossover temperatures for the onset of the screening of the orbital and spin local degrees of freedom, and for their completion. They remark the separation of orbital and spin energy scales by showing that, contrary to V₂O₃, in Sr₂RuO₄ the departure of the orbital and spin susceptibilities from a Curie-Weiss behavior happens at different temperatures.

This parallels a slow reduction with raising temperature of the spectral weight at zero frequency, without any characteristic temperature, compared to V₂O₃ in which the reduction is faster and the spectral weight reaches a plateau at a specific crossover temperature that can also be identified in all susceptibilities.

At lower temperature they define other crossover scales for the completion of the screening of these local degrees of freedom in the route towards a Fermi-liquid state, which are distinct for orbitals and spins in both V₂O₃ and Sr₂RuO₄.

The study appears technically correct, and relies on a quite consolidated technology (DFT+DMFT in its state-of-the-art implementation), however in my opinion the results themselves do not add a lot of information to the ongoing investigation of the so called Hund's metals, and their distinction from more "traditionally" known Mott-correlated metals.

Indeed some weakening points of the present study are:

- positing that the phenomenology found in V₂O₃ and Sr₂RuO₄ can be used to single out general features of Mott's and Hund's metals.

Indeed no attempt is done in disentangling the outcome of specific material features from the supposedly general phenomenology (in particular the temperature dependence of the spectral weight at zero frequency can be expected to also depend on the initial band structure and not only on the general dichotomy between Mott and Hund physics as proposed here).

If one has to use realistic simulations of materials to highlight commonalities between systems to be ascribed to one of these two kinds of physics, it is necessary to show more than two materials. Rather a series of (reasonably different) materials of each kind (Mott and Hund), so to highlight what is common to each of the two classes and distinguishes from the other, and what instead is material-dependent within each class. As one would do with experiments.

As such this is a study of two materials, and all extracted information is not a deduction on the general phenomenology, but rather a confirmation of possibly expected physics from previous studies (among which the same authors previous publications Ref. 7, 9, 23 and 26, but also much of the previous literature on the subject, summarized i.e. in Ref. 6).

- no insight is given on what makes a material of one or the other kind. That is, even admitting that V₂O₃ is of the Mott kind and Sr₂RuO₄ is of the Hund's kind, what in their composition determines that? No attempt is made in going beyond the statement that since they are supposedly archetypes of the two classes, one different feature in their physics confirms the

assumption.

- the fact that the defining features distinguishing Hund from Mott materials are found in ranges of temperatures well above those (below room temperature) where this distinction was born (pnictides, ruthenates, etc.) and where it would be useful. The mentioned findings have a conceptual applicability indeed, but restricted to a limited theoretical framework, that can hardly connect directly to experiments.

For these reasons I think that this paper does not match the requirement of having significant enough conclusions and of being of interest to a large community and I think it is more suitable for a more community-specific journal.

Reviewer #2 (Remarks to the Author):

The manuscript by Deng et al. "Signatures of Mottness and Hundness in archetypal correlated metals" tries to show difference between doped Mott insulators (e.g. V₂O₃) and so called Hund metals (Sr₂RuO₄). Both types of systems are strongly correlated multiorbital systems which requires LDA+DMFT to treat the physical problem adequately.

The major claim of the manuscript is that upon temperature increase for Mott systems quasiparticle peak is washed out completely and kind of "pseudogap" appears at the Fermi level. While for Hund metals upon temperature increase quasiparticle peak stays at the Fermi level but is strongly smeared by temperature.

In principle, the manuscript is novel enough and would be of interest for the "strongly correlated" community. However best validation of such type of theoretical statements is experiment!

The systems considered are long time under research and there must be lots of photoemission data available even for high temperatures (since melting point e.g. for V₂O₃ is 1967K). Is there any signatures of such behavior for Mott systems or Hund metals in the photoemission experiments? or may by thermodynamics? Although "wiggling" in the entropy presented by authors and interpreted as some kind of energy scales is questionable. The entropy calculation as shown in the Ref. 50 is quite complicated numerically.

Another immediately arising question is why authors have chosen Sr₂RuO₄ as a Hund metal prototype? Why not iron based superconductors? The later systems were named by some of the authors as Hund metals...

The paper could be published in the Nature Communications if authors provide comparison of their PDOS with photoemission experiments at different temperatures and such comparison proves their theoretical statement. That should not take much space in the body of the paper.

Reply to Referee #1:

[Font code: *black and italic: referee's comments; blue: our reply;*]

We thank the referee for her/his constructive suggestions and criticism that helped us to substantiate our results and to expand the scope of our work.

This article relates DFT+DMFT simulations of two materials, V2O3 and Sr2RuO4, which are taken as archetypes for Mott and Hund's correlated electronic physics in metals.

The article bases the simulations on parameters (mainly interaction couplings U and J) estimated in previous calculations from the same authors Ref. 23 and 26. Presented results focus on the temperature dependence of local spectral functions, charge, spin and orbital local susceptibilities, and impurity entropy. The authors highlight crossover temperatures for the onset of the screening of the orbital and spin local degrees of freedom, and for their completion. They remark the separation of orbital and spin energy scales by showing that, contrary to V2O3, in Sr2RuO4 the departure of the orbital and spin susceptibilities from a Curie-Weiss behavior happens at different temperatures.

This parallels a slow reduction with raising temperature of the spectral weight at zero frequency, without any characteristic temperature, compared to V2O3 in which the reduction is faster and the spectral weight reaches a plateau at a specific crossover temperature that can also be identified in all susceptibilities.

At lower temperature they define other crossover scales for the completion of the screening of these local degrees of freedom in the route towards a Fermi-liquid state, which are distinct for orbitals and spins in both V2O3 and Sr2RuO4.

The study appears technically correct and relies on a quite consolidated technology (DFT+DMFT in its state-of-the art implementation), however in my opinion the results themselves do not add a lot of information to the ongoing investigation of the so-called Hund's metals, and their distinction from more "traditionally" known Mott-correlated metals.

We thank the referee for her/his recognition of the fundamental physics explained in our manuscript. However, we respectively disagree with the comment that "*the results themselves do not add a lot of information to ...*". To the best of our knowledge, our study reveals the two distinct routes of screening from the atomic degrees of freedom towards the emerging quasiparticles, guided either by Mott or by Hund physics. We introduce the new concept of the onset and the completion of spin and orbital screening and emphasize the differing features observed in the onset scales of screening and in the correlated spectra of Mott and Hund systems, respectively, which serve as clear, novel signatures to distinguish different origins of strong correlations in various materials. We believe our study has shed light on the ongoing intensive study of Hund materials, and also deepen the understanding of Mott physics in multi-orbital materials.

Indeed, some weakening points of the present study are:

- positing that the phenomenology found in V2O3 and Sr2RuO4 can be used to single out general features of Mott's and Hund's metals.

Indeed no attempt is done in disentangling the outcome of specific material features from the supposedly general phenomenology (in particular the temperature dependence of the spectral weight at zero frequency can be expected to also depend on the initial band structure and not only on the general dichotomy between Mott and Hund physics as proposed here).

If one has to use realistic simulations of materials to highlight commonalities between systems to be ascribed to one of these two kinds of physics, it is necessary to show more than two materials. Rather a series of (reasonably different) materials of each kind (Mott and Hund), so to highlight what is common to each of the two classes and distinguishes from the other, and what instead is material-dependent within each class. As one would do with experiments.

We chose V_2O_3 and Sr_2RuO_4 as prototype materials that can be utilized to demonstrate our insights on originations of correlations and their consequences. This is based on our knowledge that these two materials have been intensively studied for decades with abundant measurements and analysis, and we believe that a consensus on their underlying physics has been reached in the community, as shown by the reports cited in our manuscript.

We understand the concern of the referee and completely agree that it is necessary and helpful ***to disentangle the outcome of specific material features from the supposedly general phenomenology.*** We thank the referee for her/his suggestions to expand the studies to other materials, which we think is quite reasonable. However, to study a set of correlated materials, identify their nature of correlations and provide sound first principle descriptions with the capability to consistently explain/justify available experimental measurements, requires not only computationally expensive and time-consuming simulations but also a significant load of theoretical investigations. Our current work benefits a lot from our previous successful studies on V_2O_3 and Sr_2RuO_4 in the past few years. And continuing such studies for other materials is a somewhat iterative or repetitive approach towards disentangling Mott and Hund features, and one may still question to which extent the materials specifications can be regarded as not important.

To circumvent the significant amount of work needed for additional realistic materials, we decided to follow a different, but in our view, more compelling procedure: we deduce the main signatures from a model Hamiltonian. This is important, as in real materials Mott and Hund physics is present in different amounts, which is hard to disentangle. Furthermore, band structure effects, crystal fields and other solid-state details might indeed impact the outcomes, as noted by the referee. Within a simple model study, in contrast, we can prove that these material-dependent details are not essential to the main physics by deriving the phenomenology associated with various parameter regimes. In addition, by introducing a model, we are in effect introducing a "series" of materials. This is in our opinion in the best tradition of solid state theory, which has always combined insights from model Hamiltonians and real materials.

We have abstracted the essence of Mottness and Hundness in a simplified Hubbard-Hund model Hamiltonian, which contains both a Hubbard U and a Hund's rule interaction J , but no material-specific details, such as the band structure. The two parameters can be varied and controlled independently in the model. For appropriate parameter choices we observe the same Mott and Hund signatures as in the archetypal materials, but here we can clearly trace back their origin to large effects of U or J , respectively, thus establishing a material-independent, general phenomenology which is broadly applicable to different materials. The model Hamiltonian was solved with DMFT and the Numerical Renormalization Group (NRG), an efficient multi-band impurity solver which was only recently successfully applied to treat the

three-band Hubbard-Hund model. The model Hamiltonian study has now been included in the main paper with an additional figure (Fig. 4 in the main text).

As such this is a study of two materials, and all extracted information is not a deduction on the general phenomenology, but rather a confirmation of possibly expected physics from previous studies (among which the same authors previous publications Ref. 7, 9, 23 and 26, but also much of the previous literature on the subject, summarized i.e. in Ref. 6).

As described above, we are now able to deduce the general phenomenology from the model, in addition to the induction from the archetypal materials. We believe that the substantial additions to the new version of the paper satisfies this criticism.

- no insight is given on what makes a material of one or the other kind. That is, even admitting that V2O3 is of the Mott kind and Sr2RuO4 is of the Hund's kind, what in their composition determines that? No attempt is made in going beyond the statement that since they are supposedly archetypes of the two classes, one different feature in their physics confirms the assumption.

This is a good question. It is indeed difficult to tell simply from its *composition* if a material is of Mott kind or Hund kind. As noted in our manuscript, from the perspective of an atomic picture, V2O3 and Sr2RuO4 are quite similar with 2 electrons (holes) in 3 t2g shells, however, we argue that they have different effective U values and hence are quite different in their distance to the Mott state and thus in the nature of their correlations. So far, in addition to the experimental investigations that map the composition/structure phase diagram, first principle methods that capture dominating correlations effects, such as the DFT+DMFT method adopted in our manuscript, are viable options to answer that question. Indeed, this question itself was the motivation to reveal possible signatures of different kinds of correlations in our manuscript. And we believe that the local quantities which we have studied provide these signatures and give insights on the origin of correlations - together with the model Hamiltonian study - as explained in the main text. These insights can now serve as guidance for further theoretical studies as well as experimental measurements.

In addition, in the Hubbard-Hund model study we are able to obtain a generic phase diagram in which the interaction U and J can be controlled separately, thus the physical consequence of the proximity to the Mott line can be studied directly. We add the generic phase diagram of the Hubbard-Hund model in Fig.4(a) of the main paper. There we explicitly demonstrate that when materials are close to the Mott transition line, they exhibit the Mott physics we deduced from V2O3, and when they are far away, they are Hund metals and behave as Sr2RuO4. This stresses the generality of the different roles of Mottness and Hundness in determining the behavior of correlations and their manifestations in local quantities.

- the fact that the defining features distinguishing Hund from Mott materials are found in ranges of temperatures well above those (below room temperature) where this distinction was born (pnictides, ruthenates, etc.) and where it would be useful. The mentioned findings have a conceptual applicability indeed, but restricted to a limited theoretical framework, that can hardly connect directly to experiments.

We thank the referee for her/his recognition of our findings, but respectfully disagree the comment that these findings “can hardly connect directly to experiments. Even though admittedly high temperature measurements are usually not easy, they are not impossible in the

two archetypal materials studied since both Sr_2RuO_4 and V_2O_3 have high melting points (around 2000K). Some measurements have been carried out up to rather high temperature, for example, in Ruthenates, the resistivity was measured up to 1300K. Nevertheless, measurements that can directly justify our theoretical predictions on V_2O_3 and Sr_2RuO_4 at high temperatures, for example, photoemissions, are not available as far as we know. These measurements are called for based on our findings. In addition, we note that an indirect evidence of the pseudogap opening in V_2O_3 has been in the optic measurements (L. Baldassarre et.al., PRB 77, 113107 (2008)), which shows the disappearance of the Drude peak at $\sim 600\text{K}$. The temperature scale is smaller but not far away from that identified in our calculations ($\sim 1000\text{K}$), which in our opinion is acceptable given that V_2O_3 is so close to a critical point.)

We note that for various available measurements carried out at relative low temperature (mostly below room temperature), such as specific heat coefficients, photoemission, transport and optics, our theoretical framework has been able to provide accurate descriptions of V_2O_3 and Sr_2RuO_4 , as documented in reference Deng et.al., PRL 113, 246404 (2014), 116, 256401 (2016). These successes gave us confidence that our predictions at high temperature are meaningful and could motivate experimental verifications.

We also point out, that subsequent developments will translate these temperature scales into frequency scales, and those will also reflect the ideas introduced in our manuscript and will be amenable to other experimental tests.

Further, our findings in the spectral evolution and the novel temperature scales are general, as supported the simplified model Hamiltonian study. They are not restricted to the two canonical materials and have broader implications. To connect our findings to experiments, it would be a viable route to exploit Mott/Hund's systems with lower characteristic energy scales. In these systems, the signatures we have identified would be easier to probe experimentally and can be used to gain insights on the underlying nature of correlations. One candidate could be organic/molecular conductors, where the interaction is typically small, (for example, the Mott scale T_M is a few tens of Kelvin in $\kappa\text{-(BEDTTF)}_2\text{X}$, see the schematic phase diagram in Eur. Phys. J. B 79, 383–390 (2011)).

For these reasons I think that this paper does not match the requirement of having significant enough conclusions and of being of interest to a large community and I think it is more suitable for a more community-specific journal.

We feel that with the addition of the model Hamiltonian studies, we have addressed the issues raised by the referee. With both the realistic material and the model study, we are providing new insights into fundamental concepts, Mottness and Hundness. We believe that the paper now, is of interest to the very broad community which is interested in strongly correlated materials.

Reply to Referee #2:

[Font code: *black, italic: referee's comments; blue: our reply;*]

We thank the referee for her/his constructive suggestions and comments that helped us to substantiate our results and to expand the scope of our work.

The manuscript by Deng et al. "Signatures of Mottness and Hundness in archetypal correlated metals" tries to show difference between doped Mott insulators (e.g. V2O3) and so called Hund metals (Sr2RuO4). Both types of systems are strongly correlated multiorbital systems which requires LDA+DMFT to treat the physical problem adequately.

The major claim of the manuscript is that upon temperature increase for Mott systems quasiparticle peak is washed out completely and kind of "pseudogap" appears at the Fermi level. While for Hund metals upon temperature increase quasiparticle peak stays at the Fermi level but is strongly smeared by temperature.

In principle, the manuscript is novel enough and would be of interest for the "strongly correlated" community. However best validation of such type of theoretical statements is experiment!

The systems considered are long time under research and there must be lots of photoemission data available even for high temperatures (since melting point e.g. for V2O3 is 1967K). Is there any signatures of such behavior for Mott systems or Hund metals in the photoemission experiments? or may by thermodynamics? Although "wiggling" in the entropy presented by authors and interpreted as some kind of energy scales is questionable. The entropy calculation as shown in the Ref. 50 is quite complicated numerically.

We appreciate very much the referee's recognition of the physics in our work, and her/his encouraging assessment that **the manuscript is novel enough and would be of interest for the "strongly correlated" community.**

We totally agree with the referee that the best validation of our theoretical insights should be experiments. Both materials have been heavily studied with various techniques. However most measurements were carried out at relative low temperatures, compared to the onset temperature scales we have identified. Experimentally measurements at high temperature that can be directly compared to our findings are few - as far as we are aware of - but not none. An indirect evidence of the pseudogap opening in V2O3 has been in the optic measurements (L. Baldassarre et.al., PRB 77, 113107 (2008)), which shows the disappearance of the Drude peak at ~600K. The temperature scale is smaller but not far away from that identified in our calculations (~1000K), which in our opinion is acceptable given that V2O3 is so close to a critical point.) Nevertheless, further measurements can in principle be carried out to high temperatures, and our predictions are verifiable even though admittedly high temperature transport is not easy.

As noted in this manuscript, we are able to capture low temperature properties of V2O3 and Sr2RuO4 reasonably well and our results are consistent with many measurements including integrated and angular resolved photoemission, transport, optic properties and so on (reference Deng et.al., PRL 113, 246404 (2014), 116, 256401 (2016)). These successes along with many

reports on V_2O_3 and Sr_2RuO_4 with similar techniques by other groups, gave us confidence in predictions made in this work.

We also point out, that subsequent developments will translate these temperature scales into frequency scales, and those will also reflect the ideas introduced in our manuscript and will be amenable to other experimental tests.

In this revised manuscript, to confirm that our findings are general instead of valid for specific materials only, we have abstracted the essence of Mottness and Hundness in a simplified Hubbard-Hund model Hamiltonian. This model contains both a Hubbard U and a Hund's rule interaction J , but no material-specific details, such as the band structure. The two parameters can be varied and controlled independently in the model. For appropriate parameter choices we observe the same Mott and Hund signatures as in the archetypal materials, but here we can clearly trace back their origin to large effects of U or J , respectively, thus establishing a material-independent, general phenomenology which is broadly applicable to different materials.

Supported by both realistic materials and the simplified model that explained above, we believe that our findings in the spectral evolution and the novel temperature scales are general, as supported the simplified model Hamiltonian study. They are not restricted to the two canonical materials and have broader implications. To connect our findings to experiments, it would be useful to exploit Mott/Hund's systems with lower characteristic energy scales. In these systems, the signatures we have identified would be easier to probe experimentally and can be used to gain insights on the underlying nature of correlations. One candidate might be organic/molecular conductors, where the interaction is typically small, (for example, the Mott scale T_M is a few tens of Kelvin in κ -(BEDTTTF) $_2$ X, see the schematic phase diagram in Eur. Phys. J. B 79, 383–390 (2011)).

Another immediately arising question is why authors have chosen Sr_2RuO_4 as a Hund metal prototype? Why not iron based superconductors? The later systems were named by some of the authors as Hund metals..

Thanks for this insightful question. The reason for our choice of ruthenates rather than pnictides to illustrate the ideas of Hundness are two manifolds: i) In Sr_2RuO_4 the large crystal field splitting between e_g and t_{2g} orbitals permits the DMFT treatment to only three t_{2g} orbitals, which is computationally much less expensive than that for pnictides, which requires a treatment for all five d orbitals. This is especially true when the full rotationally invariant Hund's coupling is included as is done in the current study. ii) there are considerable debates on whether iron pnictides are Hund metals. Instead of Hund metals, many think of them as close to a Mott transition, while others take them as weakly correlated as well. Ruthenates are less controversial and can be taken as canonical Hund metal in our opinion. We hope that, including these materials together with a model Hamiltonian study, our paper will serve as a basis for a future investigation which will help quantifying the level of Hundness and Mottness in the iron pnictides.

The paper could be published in the Nature Communications if authors provide comparison of their PDOS with photoemission experiments at different temperatures and such comparison proves their theoretical statement. That should not take much space in the body of the paper.

As explained above, the experiments have been done in a very limited range of temperatures that unfortunately it is not possible to do a meaningful comparison at the present stage. We

hope that our work, which can be viewed as a novel framework for thinking about the physics of these materials, will provide incentives to carry out future experiments, which will make this comparison possible.

Reviewers' comments:

Reviewer #1 (Remarks to the Author):

The authors have resubmitted their article and it is to be appreciated that substantial supporting work and rewriting has been put into this resubmission, taking seriously into account the referees objections.

I agree with the authors that the model analysis is the best way to answer my main concern of disentangling which features of the realistic simulations for V₂O₃ and Sr₂RuO₄ are actually "archetypal", as the title states.

My main suspicion is that these difference are only quantitative and not qualitative as the authors imply.

Two features are highlighted, that would be qualitatively different between Mott and Hund systems:

- 1) the quasiparticle peak dissolving into a "pseudogap" above a given high temperature T_M (in Mott systems) instead of a slow decay of the peak with no characteristic temperature (in Hund's systems)
- 2) the orbital and spin susceptibilities departing from Curie behavior below distinct temperatures (for Hund's systems), instead than below a common temperature (for Mott systems).

Point 1 is subtle, because of the overlap of quasiparticle-like (or what remains of these at high T) and incoherent excitations, both contributing to the spectral weight at $\omega=0$, and the doubt remains that at a much higher temperature also the Hund system would have a plateau in this quantity, as the Mott system does. The sharpness of the crossover could possibly be a distinctive feature as the authors highlight, but the Hund's decay might look much sharper on a larger temperature scale.

I second that typically near a Mott transition Hubbard bands will be more well-separated than far from it, thus leaving a clearer signature of a sharp crossover in the quasiparticle peak, but this seems to me a feature linked to the different proximity to a Mott transition, but not a lot to do with Hund's physics (i.e. it might be happening also in a system with a single electron per site, or even in a one band model, with different proximity to a Mott transition).

Point 2 seems to me more clear cut to make a case of Hund vs Mott physics (albeit - I remain of my opinion - hardly useful in an experimental setting due to the very high temperatures). On this the authors claim the analogy between the data in Fig. 3a and Fig. 3c with the data in Fig. 4e and 4g. The claim is that both orbital and spin $T^*\chi(0)$ reach a plateau at the same temperature T_M in V₂O₃, and so they do in the model (albeit after a very high "bump", not seen in the realistic simulation). In Sr₂RuO₄ instead the same quantity for spins reaches a plateau at a temperature in which the orbital one is far from flat, and the authors claim that this is also happening in the model. However looking at the model data Fig. 4g the reader is left with the doubt that the solid line has not actually reached the Curie behaviour at the point where the plot is cut, but after having peaked at $T^{\wedge}_{\text{onset_spin}}$ would bend down and reach an actual plateau at a lower value for much higher temperatures. In other words one is left with the doubt that data in Fig 4g is the low- T part of a "stretched" version of the data in Fig. 4e. If this were true the main smoking gun for a qualitative difference between Mott systems and Hunds systems would be disproved.

The authors do convincingly discuss the origin of the model "bump" and why it could fail to be realized in the simulation for V₂O₃. But the same reasons could be the origin of the difference between V₂O₃ and Sr₂RuO₄, since the two compounds have a different crystal field.

I thus invite the authors to show the data in Fig. 4g for higher temperatures, in order to show if the $T^*\chi(0)$ for spins bends down and reaches a plateau or not, and thus confirm or disprove the smoking gun evidence for a qualitative difference with Fig. 4e.

A mere quantitative difference in the “archetypal” features between Mott and Hund’s systems, as defined by the authors would not make the case for an article in Nature Communications, in my opinion.

Reply to Reviewer #1:

[Font code: *black and italic: referee's comments*; blue: our reply;]

The authors have resubmitted their article and it is to be appreciated that substantial supporting work and rewriting has been put into this resubmission, taking seriously into account the referees objections.

I agree with the authors that the model analysis is the best way to answer my main concern of disentangling which features of the realistic simulations for V₂O₃ and Sr₂RuO₄ are actually “archetypal”, as the title states.

We are glad that Reviewer #1 recognizes the effort we put into revising our article, and more importantly, the strategy that we took to solve a fundamental problem in condensed matter physics: how to distinguish two distinct roads for correlating compounds – Mott physics and Hund physics.

Now our paper has both first-principles calculations for two real materials, and a simplified model Hamiltonian calculation where the proximity to the Mott transition – and therefore Hund and Mott features – can be ascertained without any doubt [Fig.4(a) in main paper].

My main suspicion is that these difference are only quantitative and not qualitative as the authors imply.

While we understand the reviewers concern in principle, we believe that in the present case the quantitative differences are so strong that they lead to qualitatively different behavior. Let us elaborate:

Strictly speaking, in the absence of a broken symmetry and at finite temperatures, all states of matter are adiabatically connected, and in this sense do not differ “qualitatively”, only “quantitatively”. From this perspective, a paramagnetic Mott insulator, a Hund metal and a good metal would all differ only “quantitatively”, as they do not break symmetries and can be adiabatically connected to each other. The same perspective, though, would draw no qualitative distinction between a liquid and a gas (no broken symmetries, adiabatically connected). However, they exhibit a quantitative difference – liquids have much higher density than fluids – with such strong consequences that qualitatively different behavior (e.g. regarding the compressibility) does emerge.

Similarly, we contend that multi-orbital doped Mott insulators and Hund metals exhibit a quantitative difference so important that it leads to qualitatively different behavior. The quantitative difference becomes apparent when comparing the lowest energy scale, E_{atomic} , of all the bare atomic excitations that constitute the Hubbard bands, to the onset scale of orbital screening, $T_{\text{orb}}^{\text{onset}}$, corresponding to the overall width of the quasiparticle peak: **Mott systems have $E_{\text{atomic}} \gg T_{\text{orb}}^{\text{onset}}$, whereas Hund systems have $E_{\text{atomic}} \approx T_{\text{orb}}^{\text{onset}}$. This leads to qualitative different temperature-dependent signatures:** with increasing temperature Mott systems develop a pseudogap at temperatures well below E_{atomic} , whereas Hund systems do not, since the emergence of a pseudogap goes hand in hand with the breakdown of spin and orbital screening and hence requires $T > T_{\text{orb}}^{\text{onset}}$. This point, which we had not formulated sufficiently clearly in the previous version of the paper, is now addressed prominently on p.2 in the revised manuscript, and again in the concluding paragraph.

Moreover, for Hund metals the onset scales for orbital and spin screening are well separated, opening a new large energy regime in Hund metals where intriguing Hund physics occurs: large, almost unscreened spins (leading to large Curie-like spin susceptibilities) are coupled to screened orbital degrees of freedom. In contrast, in Mott systems, the onset of screening with decreasing temperature occurs simultaneously in both spin and orbital channels with the formation of a quasiparticle peak inside a pseudogap.

In our paper, we introduced the new concept of the orbital and spin screening onset scales to distinguish Hund and Mott materials.

We will now answer all individual concerns brought up by Reviewer #1:

Two features are highlighted, that would be qualitatively different between Mott and Hund systems:

1) the quasiparticle peak dissolving into a “pseudogap” above a given high temperature T_M (in Mott systems) instead of a slow decay of the peak with no characteristic temperature (in Hund’s systems).

Yes, this is correct. This is one insight that we provide: the crossover region which marks the onset of the quasiparticle buildup is much broader in the Hund than the Mott case. In the latter, the crossover regime shrinks down to a sharp feature at a temperature scale T_M . This is also very clear in the model Hamiltonian study [compare insets of Figure 4(c) and (d)].

To emphasize and clarify this idea even more, we decided to add two cartoons next to the real material calculations [compare insets of Figure 1(d) and (f)]. In particular, we want to make clear that *also* the real materials, *independently* of band structure details, show the features illustrated in the cartoons: when temperature is decreased, in the Mott case (vanadium oxide) the coherence peak develops rather suddenly from within a broad minimum (pseudogap), whereas in the Hund case (ruthenate), the sharp coherence peak develops very gradually from a broad maximum.

2) the orbital and spin susceptibilities departing from Curie behavior below distinct temperatures (for Hund’s systems), instead than below a common temperature (for Mott systems).

Yes, this is correct.

The referee requested to see higher temperatures for the Hund orbital and spin susceptibilities in the model. We provide that additional information in our revised manuscript in the inset of Fig. 4(g) by enlarging the temperature range up to $T=1$. We point out, that this does *not* modify our conclusions (see further below for details regarding the Reviewers point 2).

Point 1 is subtle, because of the overlap of quasiparticle-like (or what remains of these at high T) and incoherent excitations, both contributing to the spectral weight at $\omega=0$, and the doubt remains that at a much higher temperature also the Hund system would have a plateau in this quantity, as the Mott system does.

To counter the referee's doubt, we have for completeness added a new curve for $T=1$ in black in Fig.4(d), and even then a resonance peak persists.

However, note that this temperature is already larger than the energy scale of the lowest atomic excitations, E_{atomic} , which in the model Hamiltonian study is estimated in terms of a Hubbard I analysis by $E_{\text{atomic}} = 0.5U - J = 0.5$. We would like to argue that such high temperatures do not merit a detailed discussion in the context of the present paper, and therefore mainly focus on temperatures below the bare atomic excitation scales in our revised manuscript.

We summarize: the spectral weight at the Fermi level does *not* reach a plateau (pseudogap) for Hund systems (not even at $T=1 > E_{\text{atomic}}$). This persistence of the resonance peak for Hund systems stands in stark contrast to Mott systems, where a pseudogap already develops at moderate temperatures, well below bare atomic energy scales. These ideas are very clearly supported in the model study, as well as in the ab-initio calculations of the prototypical materials.

The sharpness of the crossover could possibly be a distinctive feature as the authors highlight, but the Hund’s decay might look much sharper on a larger temperature scale.

We show the crossover in $D(i \omega_0)$ in the insets of Figure 4(c) and (d) for the Mott and Hund case respectively, now for an enlarged energy window up to $T=1$, following the request of the Reviewer. Still, the Hund decay does not look sharp on this enlarged scale, as a pseudogap can *not* be reached at moderate U (as explicated already above), even at temperatures exceeding E_{atomic} . Moreover, $D(i \omega_0)$ is always larger in the Hund metal case than in the Mott case.

For real materials we have to use the actual temperature scales, and also here, the crossovers in $D(i \omega_0)$ are very different [compare Fig.1(a) and (b)] in the Mott and Hund system.

Thus, the sharpness of the crossover IS a distinctive qualitative feature.

It is true that in principle one can make two crossovers look alike by plotting temperature in units of the crossover scale – however, this is meaningful only if that scale is well separated from other energy scales in the problem. That is not the case for Hund systems, where $T_{\text{orb}}^{\text{onset}}$ is comparable to E_{atomic} , and therefore it is not meaningful to consider temperatures much above $T_{\text{orb}}^{\text{onset}} \approx E_{\text{atomic}}$.

Of course, in the model Hamiltonian calculations, we can change the ratio of bare atomic and screening scales continuously from order 1 to very large values by changing U and J . However, it is one of our main results that features present in the ruthenates are similar to those occurring *far* from the MIT in the model Hamiltonian study and are thus "Hund metal features", while features present in V_2O_3 are found *close* to the MIT and are thus "Mott features".

I second that typically near a Mott transition Hubbard bands will be more well-separated than far from it, thus leaving a clearer signature of a sharp crossover in the quasiparticle peak, but this seems to me a feature linked to the different proximity to a Mott transition, but not a lot to do with Hund's physics (i.e. it might be happening also in a system with a single electron per site, or even in a one band model, with different proximity to a Mott transition).

The Reviewer's statement that typically near a Mott transition Hubbard bands will be more well-separated than far from it, is of course correct. But this is not the main issue that we address.

There is an ongoing discussion about the question if and (if yes) which materials are correlated through Hund physics and how these can be distinguished from Mott materials. Thus we want to clarify in our manuscript *where* to place the real materials in the phase diagram and identify the essential distinct features that emerge for Mott and Hund materials, respectively.

By comparing realistic and model calculations, we have shown where to place two archetypal materials in the phase diagram and therefore assign features of Mott and Hund physics. These insights allow us to clarify that ruthenates are indeed Hund materials: we know now that they are far from the MIT *because* they have – similar to the model Hund system – a stronger overlap of Hubbard bands and a slow decay of $D(i \omega_0)$. In contrast, V_2O_3 has to be close to the MIT *because* it has a pseudogap at high T similar to the model Mott system.

Of course, appreciating this feature is trivial once one has accepted the insights of this paper (which shows the importance of this paper!). In this respect we hypothesize in our manuscript that a gap/pseudogap regime appearing in the local spectra when the coherence resonance is destroyed is a defining signature of Mott physics in *general* situations and test it in the metallic phase of V_2O_3 (p.3 of our revised manuscript).

Therefore Reviewer #1 is correct: the sharpness of the crossover in $D(i \omega_0)$ *is* indeed linked to the different proximity to a Mott transition in *general* situations (and yes, a sharp crossover is also found near the MIT in one-band Hubbard systems). Conversely, we here argue that *strongly* correlated materials which do not show a sharp crossover $D(i \omega_0)$ are consequently not Mott materials.

Of course, it is a valid question to ask (as Reviewer #1 does) in which way exactly a Hund system is then different from a simple-one band system that is *far* from the MIT. First, the latter is weakly correlated, whereas a Hund system is strongly correlated, despite being far away from

a Mott transition. This can be shown in detail, by studying the quasiparticle weight, Z , which is large for a one-band model far from the MIT but small for Hund systems (as will be shown in a separate publication). Second, a one-band model does not involve any orbital degrees of freedom, whereas a Hund metal does. Far away from the MIT, this leads to a large energy window in Hund metals involving screened orbital degrees of freedom coupled to large, almost unscreened spins. The latter show Curie-like behavior whereas the former do not, as shown in Figs. 3(c) and Fig. 4(e).

Furthermore, the behavior of $D(i \omega_0)$ and these Hund signatures make it possible to clearly distinguish also *multi-orbital* Mott and Hund systems, *although* they are *both* strongly correlated.

Point 2 seems to me more clear-cut to make a case of Hund vs Mott physics (albeit - I remain of my opinion - hardly useful in an experimental setting due to the very high temperatures).

We argued that the fact that something has not been measured does not mean that it cannot be measured. We even pointed out some materials where those measurements could be carried out easily, such as in the organic semiconductors where the energy scales are much smaller.

On this the authors claim the analogy between the data in Fig. 3a and Fig. 3c with the data in Fig. 4e and 4g. The claim is that both orbital and spin $T^ \chi(0)$ reach a plateau at the same temperature T_M in V_2O_3 , and so they do in the model (albeit after a very high “bump”, not seen in the realistic simulation).*

Yes. This is a main point.

*In Sr_2RuO_4 instead the same quantity for spins reaches a plateau at a temperature in which the orbital one is far from flat, and the authors claim that this is also happening in the model. However looking at the model data Fig. 4g the reader is left with the doubt that the solid line has **not actually reached the Curie behaviour** at the point where the plot is cut, but after having peaked at T_{spin}^{onset} would **bend down and reach an actual plateau at a lower value for much higher temperatures**. In other words one is left with the doubt that data in Fig 4g is the low- T part of a “**stretched**” version of the data in Fig. 4e. If this were true the main smoking gun for a qualitative difference between Mott systems and Hunds systems would be disproved.*

As requested by the reviewer, we show additional data for the Hund system in the new inset of Fig. 4(g) going up to high temperatures ($T=1$, which is already larger than the atomic excitation energy of $E_{atomic} = 0.5$). In this range the product of temperature and *orbital* susceptibility, $T\chi_{orb}(0)$, keeps increasing with increasing temperature. In contrast, the product of temperature and spin susceptibility, (i) reaches a very broad plateau-like maximum around $T=0.5$ and reduces, but only slightly, above $T=0.5$ with increasing temperature. (ii) However, it never reaches an actual plateau at a lower value as the temperature continues to increase, because Hund metals do not form a pseudogap.

Notice that for temperatures above approximately $E_{atomic}=0.5$, atomic physics come into play in the model, affecting the spin susceptibility. This regime is not yet reached in the realistic calculations.

To provide a meaningful comparison of model and realistic calculations, we have retained, for the model results shown in the main figures Fig.4(g) and (h), a temperature range that does not exceed E_{atomic} .

Summarizing this discussion: following the request of the Reviewer we have extended the temperature range in the inset of Fig.4(g), but our conclusions remain unchanged: the behavior found in the model, for the Mott and Hund parameters, is similar to the behavior of the two archetypal materials.

The authors do convincingly discuss the origin of the model “bump” and why it could fail to be realized in the simulation for V_2O_3 . But the same reasons could be the origin of the difference between V_2O_3 and Sr_2RuO_4 , since the two compounds have a different crystal field.

We are sure that crystal field effects are not relevant for this issue: Figs.1 (c) and (d) for V_2O_3 show a resonance that develops from a pseudogap, for *both* orbitals. Fig.1 (e) and (f) for Sr_2RuO_4 show a resonance that does not develop from a pseudogap but an incoherent broad background, for *both* orbitals. This shows very clearly that band structure details cannot explain the main difference between V_2O_3 and Sr_2RuO_4 . This conclusion is confirmed by the similarity between most of the properties (particularly for the spectral functions) of the degenerate model Hamiltonian study and the realistic calculations.

*I thus invite the authors to show the data in Fig. 4g for higher temperatures, in order to show if the $T^*Chi(0)$ for spins bends down and reaches a plateau or not, and thus confirm or disprove the smoking gun evidence for a qualitative difference with Fig. 4e.*

We have complied with the Reviewer's request and enlarged the temperature window in the inset of Fig. 4(g) for the Hund system. As explained above, $T_{\chi_{spin}(0)}$ does NOT reach an (additional large-temperature) plateau even up to temperatures well beyond E_{atomic} , in contrast to the Mott system (and we insist that exploring the regime $T \gg E_{atomic}$ is not meaningful). We thus confirm the smoking gun evidence for a qualitative difference between Figs. 4(g) and 4(e).

A mere quantitative difference in the "archetypal" features between Mott and Hund's systems, as defined by the authors would not make the case for an article in Nature Communications, in my opinion.

We believe that the additional data and the detailed explanations we provide in our revised manuscript and in this reply letter leave no doubt that the archetypal features we revealed to distinguish correlations of Mott and Hund type are *qualitative*.

Although at first glance Mott and Hund systems might seem to differ only quantitatively, what matters for qualitative behavior is the relation of the (lowest) bare atomic excitation energies and the screening scales of the quasiparticles: $E_{atomic} \gg T_{orb}^{onset}$ for Mott systems vs. $E_{atomic} \approx T_{orb}^{onset}$ for Hund systems. This difference ultimately leads to the emergence of qualitatively different signatures of Mottness and Hundness. These signatures are present in both model Hamiltonians and material specific calculations.

We stress that the "Hund road" is a new and hot subject with significant implications and far-reaching insights for the broad field of strongly correlated materials including high temperature superconductors.

It is very general and applies to many more multi-orbital materials than the "Mott road", in the sense that Mott physics requires adjusting parameters to be *close* to a Mott transition, while Hund physics happens *close and far* from the Mott transition.

In Mott materials, i.e. *close* to a Mott transition, Hund features such as spin-orbital separation are present, if at all, only at very low temperatures or energies (in the completion of screening), while at intermediate and high temperatures Mott (atomic-like) physics dominates. In Hund compounds, i.e. *far* from the Mott transition, Hund physics (spin-orbital separation) occurs at higher energy scales and governs a broad range of energies including the intermediate and high energy regime, while Mott physics (i.e. a gap or pseudogap) is absent.

We believe that our paper elucidates in a very clear fashion a physics point which is very controversial at the present time, because the community has so far not been able to distinguish the different physical manifestations of Mott and Hund physics. The publication of this paper will make a major contribution toward clarifying this issue.

Reviewers' comments:

Reviewer #1 (Remarks to the Author):

The data added by the authors (i.e. the larger temperature range shown in the insets of Fig. 4g) upon my request unfortunately prove my point, that there is no qualitative difference between the two chosen cases opposing putatively an "archetypical" Hund vs Mott metals. The difference is quantitative instead between the two presented systems (and the two corresponding cases in the model).

This motivates the authors in arguing that strong quantitative differences can be regarded as quantitative, by using the examples of metal vs Mott insulator or even liquid vs gas. However these examples are misleading in my opinion because they concern cases where there are zones of the phase diagrams where the changes are sudden between the two phases (a phase transition, or a sharp crossover). Correspondingly the differences between the two phases are very sharp in that region. This is in my opinion what justifies the distinction. In other parts of the phase diagram where the adiabatic connection results in a very smooth evolution of the physical properties typically this distinction is dropped.

What the authors have done in this work, in my opinion, is taking two systems, both with correlations driven by U and (most importantly) by J , but one close to the *density-driven* (in their response the authors talk about "doped Mott insulators" but here both systems have $\langle N \rangle = 2$) Mott transition and the other far from it. Both could be called "Hund metals", and show the remarkable features (like the spin-orbital separation in the screening at low temperatures) of such metals compared to normal ones. Each one displays these features to a different degree, according to the different typical temperature scale of each. On top of that, one of the two shows also the signs of the nearby interaction-driven Mott transition (which can be identified with the authors' definition of large E_{atomic} compared to the screening scale).

It thus appears unnecessary to contrast Hund and Mott metals, as hereby defined.

I agree with all the rest of the analysis that the authors include in their rebuttal: that Hund's metals provide a new road for correlations even far from the Mott transition, that this physics is much more common than finding materials close to a density-driven Mott transition like V_2O_3 , and that this is very interesting. But all this has been analyzed already in the literature (Refs. 4 to 15 and others, many by the same group).

In summary in this work the authors report a honest study of two Hund's metals, one of which close to the Mott transition, but contrasting Hund and Mott as separate paradigms is misleading, in my opinion. I think this work should not be published with this misleading message.

Reviewer #3 (Remarks to the Author):

I have read with considerable interest the revised manuscript(s) by Xiaoyu Deng et al., "Signatures of Mottness and Hundness in archetypical correlated metals", as well as all the enclosed, extensive correspondence with the Referees.

I agree with the Authors about the fact that their manuscript is focusing on a very important and vividly debated issue in the cutting-edge research on condensed matter systems, i.e., the rigorous identification of the physical properties of the so-called Hund's metals. In fact, even after (about) ten years after the introduction of this concept, triggered by DMFT studies on the strong Hund's coupling (J) dependences in three or five orbital systems, a precise classification of those phenomena, to be regarded as a decisive fingerprint of the Hund's metal nature of a given compound is still lacking.

Far from being a mere academic problem, this discussion is important also for improving, in a significant way, the interpretation of present and future experimental results and for guiding the corresponding theoretical analysis.

Hence, I do not think that making a direct contact with one or more specific experiments might be considered as a necessary requirement for the publication of this work, since the major goal, here, is to individuate one or more "smoking guns" properties to identify, unambiguously, the Hund's metal physics.

In this respect, however, I must say that I understand, to a certain extent, the concerns expressed by the first Referee in all her/his reports. The reason for this is related to a somewhat disordered way the ideas and the data are discussed in the (now revised) manuscript and to the overall impression, left to the reader, that after looking at several data, the "smoking gun" of the Hund's metal physics is not fully provided.

As this is clearly the key point of the manuscript, I think that, in the same Authors' interest, this issue should be clarified in the most possible transparent and convincing way. This will require additional, but focused DMFT calculations, as well as specific changes in the discussions of the manuscript.

Here are the main points which have to be considered:

1) One-particle spectra: It is essential to distinguish the phenomena which can be "compatible" with a Hund's metal physics from those which can be regarded as stronger (or even univocal) indication of such physics. In this respect, I tend to agree with the first Referee that the temperature evolution of the spectral function of the HM appears really similar to those weakly correlated phase of a doped Mott insulator. This will make hard an unambiguous identification of "Hundness", simply by looking at the T-dependence of the photoemission spectroscopy data (not to speak that the latter can be also considerably affected by several features, e.g. proximity of to a Van Hove Singularities or more generically to prominent features of the DOS close to E_F , etc.). In this respect, the relation of the T-dependences with E_{atomic} (which might be regarded a more distinctive difference) does not look to me "strong" enough as a clear-cut criterium, unless this information could be also extracted, independently, from the same photoemission data set.

2) Susceptibilities (low T): The differences observed in the the orbital/charge and spin susceptibilities at low-T have been already extensively discussed, in a similar context, in a recent preprint (Ref. [12]) by some of the same Authors of this work. Hence, the low-T analysis can be not considered a completely new result and, in addition, it does not provide a sharp criterion as the high-T one, because all mentioned temperature-scales go (necessarily) to zero at the MIT, keeping a roughly constant ratio.

3) Susceptibilities (high T): From the discussions above, it is clear that the high-T results represent the newest and most relevant piece of information provided by this paper. It is true that the (onset) temperature at which the qualitative differences between the charge/orbital and spin sectors are predicted (i.e., the temperature above which a "Curie behavior" sets in) will be in some cases too high to be easily observed experimentally. However, such differences might indeed represent -in this I agree with the Authors- a more genuine hallmark of an Hund's metal behavior. Hence, I think that the Authors have a considerable interest to demonstrate, beyond any doubt, that this is indeed the case, and that the legitimate concerns of the first Referee (who suspects the possibility of interpreting even this feature, more generally, in term of a larger/smaller proximity to a MIT) can be disproved. The additional model calculations I am proposing below aim at this goal.

Finally, in the context of the appearance of a $1/T$ dependence of the susceptibilities, it is important not to forget that this behavior might be triggered also by the proximity of a flat band to the Fermi level, as it happens, e.g., in elemental Ni.

4) Real material calculations: Because of the "fundamental" nature of the subject discussed here, I regard the realistic DMFT calculations for the two prototypical materials, i.e., V₂O₃ and Sr₂RuO₄, more as a generic/testbed example for the two classes of materials. Hence, I understand that too specific material details are not (and should not) be considered.

However, in my opinion, this choice should be clarified, more explicitly, in the paper, because, otherwise, the readers could consider the DMFT calculations for the temperature dependences of the different physical quantities as a *quantitative* result, to be directly compared with experiments. This is obviously not always the case here.

For instance, as far as I understand, in the case of V₂O₃, the temperature dependence of the lattice parameters (crucial for getting a good agreement with IR-spectroscopy experiments at high-T [see Baldassarre et al., that the Authors mentioned in one their replies]) has been not explicitly considered, leading, most likely, to results progressively more metallic than in experiments, when T is increased (Note: in the IR-experiments, a low frequency downturn in the dynamic conductivity appears above 500 K, while the presented DMFT calculations with fixed (room T) bandstructure input remain somewhat "metallic" up 800-1000 K). That this piece of information is (comprehensibly) missing in the presented study should be explicitly mentioned (with a brief reference to the works, where these issues are treated), and the readers should be warned *not* to expect an overall good agreement between the calculated data and the experiments in the whole temperature range.

As the same time, I appreciated the effort already made by the Authors in supplementing their original results with a model (material free-parameter) calculations, which - I think- it was absolutely necessary for this manuscript.

On the basis of the above considerations (1)-(4), I think that further improvements of the manuscript are still necessary, along the lines suggested in the following: (i) By making the identification of the smoking gun(s) for the Hundness more clear-cut and (ii) by discussing more explicitly the criteria which can be used just as an hint of compatibility with the Hund's Metal/Mott insulating physics w.r.t. to those providing a more stringent indication; (iii) the Authors should better clarify the differences with the analysis made in Ref. [12] ; (iv) They must better highlight the scope of the DFT+DMFT calculations for V₂O₃ and Sr₂RuO₄, warning the reader about the missing ingredients for a realistic comparison with experiments [cf. Baldassarre et al. and/or similar], and discuss realistic effects which could potentially hide/change the main features discussed in their manuscript.

As for (i), a convincing proof of the main message of the paper will be showing *model* calculations (of all physical quantities considered and, above all, of their temperature dependences) for cases with a *indisputable* Mott-nature at different "distances" from the Mott-MIT.

Possible choices: the correlated metallic phase of the half-filled Hubbard model for $U \ll U_c$, and/or of the doped single-orbital model for $U > U_c$, and/or three-orbital model where J has been put (somewhat artificially) to zero, or three (perfectly degenerate) orbital models with finite (Kanamori) J, but perfectly half-filled (where the onset of the Mott-phase is strongly favored by both the U and the J terms).

The DMFT results of (at least) one of such clear-cut cases should be critically compared to the ones already presented in the manuscript. This is particularly important for the data in the weak-to-intermediate regime, i.e., at significant "distance" from the MIT, in order to directly visualize the difference between the two situations: metallic systems relatively far from the Mott MIT vs. Hund's metal.

As for (ii), I would seriously consider (though I leave the final decision to the Authors) the idea of inverting the order of the presented results starting by the model calculations and, then, to use the results for the two prototypical materials to demonstrate how the proposed classification would work in practice. In fact, while this order might not reflect the "history" or the manuscript, it would appear much more logical for future readers.

Moreover, I think it is necessary, possibly in the very first part of the work, to present a schematic picture sketching the role of the different temperature scales in the different cases and making an explicit reference to it when discussing the subsequent numerical data and in the conclusion section. This should be supplemented by a sketchy cartoon of the microscopical processes at work in the different regimes. Further, the discussion of the "identification strength" of the different criteria should be included and/or extended in the manuscript points, where numerical data are presented.

As for (iii) and (iv), these point would essentially require an extension of the existing discussions (e.g., about the low-T calculations of the susceptibilities and their relation to Ref. [12], about the neglected material specific information, about the possible influence of specific band-structure features, like VHS, on the T-dependence of spectra and and susceptibilities, etc.).

I think that, by properly considering these points, the Authors will be able to improve considerably their presentation and to clarify some crucial part of their message and results, which would make, eventually, the paper suited for publication in Nature Communications.

PS In the course of the preparation of my report, I have received a newly revised version of the manuscript, where the Authors have significantly corrected some of their numerical results for the model hamiltonian.

As the previous results were -as far as I understand- merely originated by a technical error, and the new results fully clarify an important question posed by the first Referee, I have eliminated any further comment about this point from my report. In this respect, I just wonder whether the Authors are completely confident, now, that the residual small deviations of the static susceptibilities w.r.t. the expected high-T asymptotic can be fully ascribed to a "mixed valence" effects (s. caption of Fig. 4) rather than to numerical accuracy. Do they have some evidence for making this statement?

Detailed response to the Reviewers' comments

Color code: blue: Reviewers' comments; black: our reply; green: revisions/additions to the paper. For ease of reference, the principal changes in the main text are also typeset green in the resubmitted version. References mentioned below refer to the revised submission.

Since the revisions prompted by Reviewer #3 were rather substantial, we have taken the liberty of additionally implementing some cosmetic editorial changes (all marked in green). Most notably, we introduced section headings to better highlight the structure of our paper, and introduced a uniform color scheme for data referring to the two material systems V_2O_3 (orange) and Sr_2RuO_4 (blue).

Since our reply to Reviewer #1 will refer to our reply to Reviewer #3, we begin with the latter.

Reply to Reviewer #3

Reviewer #3 concisely summarizes why the questions addressed in our work are important: ... their manuscript is focusing on a very important and vividly debated issue in the cutting-edge research on condensed matter systems, ... a decisive fingerprint of the Hund's metal nature of a given compound is still lacking, ... this discussion is important also for improving, in a significant way, the interpretation of present and future experimental results and for guiding the corresponding theoretical analysis. However, he/she argues that the "smoking gun" of the Hund's metal physics is not fully provided, and makes four constructive suggestions for transparently clarifying this issue.

(i) Reviewer #3 suggests making the identification of smoking gun(s) for the Hundness more clear-cut, ... by showing *model* calculations (of all physical quantities considered and, above all, of their temperature dependences) for cases with a *indisputable* Mott-nature at different "distances" from the Mott-MIT. He/she suggests several possible choices for this purposes. ... The DMFT results of (at least) one of such clear-cut cases should be critically compared to the ones already presented in the manuscript. This is particularly important for the data in the weak-to-intermediate regime, i.e., at significant "distance" from the MIT, in order to directly visualize the difference between the two situations: metallic systems relatively far from the Mott MIT vs. Hund's metal.

We have implemented this suggestion for one of the choices suggested by Reviewer #3, namely the three-orbital model where J has been put (somewhat artificially) to zero. To this end, we performed additional DMFT+NRG calculations for the $J = 0$ version of our model. The corresponding results for the orbital and spin susceptibilities have been included in Figs. 5(e-h), and are discussed near the bottom of p. 7, and in the caption of Fig. 5. Comparing the $J = 0$ and $J = 1$ results, one sees that the former show no spin-orbital separation for the onset of screening, just as the $J = 1$ Mott system, but in contrast to the $J = 1$ Hund system. Thus spin-orbital separation for the onset of screening is a true fingerprint of Hund physics.

Our new $J = 0$ results for spectral functions have been included in a new section on photoemission spectra in the supplementary material. (A few sentences near the top of p. 9 of the main text draw attention to this new material.) Our spectral functions and corresponding photoemission predictions are shown in a new Fig. S-2, comparing the spectra predicted for our $J = 1$ Mott and Hund systems to corresponding spectra for $J = 0$. There we write: "The fairly small slope of the dispersion (i.e. large effective mass) for the $J = 1$

Hund system (third column) compared to the corresponding $J = 0$ weakly correlated system (fourth column) is a smoking gun difference between a true Hund system and a pure Mott system tuned far from the Mott transition – even though both have rather weak U , the former has strong correlations (induced by finite J), the latter does not.” Moreover, “the shape of the quasiparticle peak shows substructure indicative of spin-orbital separation for sizable J , but not for $J = 0$.”

(ii) Reviewer #3 advises us to discuss more explicitly the criteria which can be used just as an hint of compatibility with the Hund’s Metal/Mott insulating physics w.r.t. to those providing a more stringent indication. To this end, he suggests (but leaves the final decision to us) the idea of inverting the order of the presented results starting by the model calculations and, then, to use the results for the two prototypical materials to demonstrate how the proposed classification would work in practice. Moreover, he advises us to include a schematic picture sketching the role of the different temperature scales in the different cases, ... and a sketchy cartoon of the microscopical processes at work in the different regimes.

Stimulated by this excellent advice, we have added a new Fig. 1 and extensively revised the first two-and-a-half pages of our paper. Fig. 1 illustrates the microscopic processes involved in orbital and spin screening, and shows a schematic sketch of the behavior of four characteristic temperature scales, $T_{\text{orb}}^{\text{onset}}$, $T_{\text{spin}}^{\text{onset}}$, $T_{\text{orb}}^{\text{cmp}}$, $T_{\text{spin}}^{\text{cmp}}$, marking the onset and the completion of screening of orbital and spin degrees of freedom, respectively, as functions of the bare gap, $\Delta_b = U - 2J$, between the upper and lower Hubbard side band. We also indicate which regions of the figure correspond qualitatively to the two material systems V_2O_3 and Sr_2RuO_4 . In our discussion of Fig. 1 (near the middle of page 2) we remark: “The most striking observation is that increasing U pushes the onset scales $T_{\text{orb}}^{\text{onset}}$ and $T_{\text{spin}}^{\text{onset}}$ closer together until they essentially coincide. As a consequence, the Hund regime (small U) and the Mott regime (large U , close to the Mott transition), though adiabatically connected via a crossover regime, show dramatic differences for the temperature dependence of physical quantities (discussed below).”

We trust that Fig. #1 will greatly help readers to grasp the “big picture”. Moreover, our discussion of Fig. #1 in the second section, **Overview of results**, partly implements Reviewer #3’s suggestion to first outline the behavior of our model before addressing the two material systems. However, for the subsequent detailed presentation of our results, we have retained the original order – first material results, then model results – since reversing this order would have required completely rewriting the entire paper.

(iii) The Authors should better clarify the differences with the analysis made in Ref. [12] (this has now become Ref. [10]). This point would essentially require an extension of the existing discussions (e.g., about the low- T calculations of the susceptibilities and their relation to Ref. [12]).

We have addressed this point by simply including the following sentence in the introduction (p. 1): “Whereas Ref. [10] focused on $T = 0$, here we focus on temperature dependence.” It goes without saying that the low- T model results for the susceptibilities presented here are of course consistent with those of Ref. [10].

The only $T = 0$ data in the present paper is contained in Figs. 5(a,b). These are consistent with, but not identical to Figs. 5 and 12 of Ref. [10], respectively. However, please note that the first time we submitted this data for publication was in the second version of the *present* paper, submitted to Nature Communications in May 2018, well before we posted Ref. [10] to

the arXiv in August 2018. Therefore its inclusion here is well justified by historical priority. We have nevertheless added the remarks “(cf. Fig. 5 of Ref. [10])” and “(cf. Fig. 12 of [10])” to the captions of Fig. 5(a) and 5(b), respectively.

(iv) They must better highlight the scope of the DFT+DMFT calculations for V₂O₃ and Sr₂RuO₄, warning the reader about the missing ingredients for a realistic comparison with experiments [cf. Baldassarre et al. and/or similar], and discuss realistic effects which could potentially hide/change the main features discussed in their manuscript. Reviewer #3 had elaborated on the missing ingredients somewhat earlier in his/her report: as far as I understand, in the case of V₂O₃, the temperature dependence of the lattice parameters (crucial for getting a good agreement with IR-spectroscopy experiments at high-T [see Baldassarre et al., that the Authors mentioned in one their replies]) has been not explicitly considered That this piece of information is (comprehensibly) missing in the presented study should be explicitly mentioned (with a brief reference to the works, where these issues are treated), and the readers should be warned *not* to expect an overall good agreement between the calculated data and the experiments in the whole temperature range.

We have addressed this issue by adding the following sentences to our discussion of the real material calculations (p. 3, near top of last paragraph): “We have not taken into account the effects of the temperature-dependent changes in lattice parameters, which have been shown to be very important in materials near the Mott transition such as V₂O₃ [32]. Nevertheless, the LDA+DMFT calculations here bring a degree of realism, such as band structure and crystal field effects, which is not present in the 3HHM calculations discussed further below.”

Reviewer #3 raises one more issue: PS In the course of the preparation of my report, I have received a newly revised version of the manuscript, where the Authors have significantly corrected some of their numerical results for the model hamiltonian.

As the previous results were – as far as I understand – merely originated by a technical error, and the new results fully clarify an important question posed by the first Referee, I have eliminated any further comment about this point from my report. In this respect, I just wonder whether the Authors are completely confident, now, that the residual small deviations of the static susceptibilities w.r.t. the expected high- T asymptotic can be fully ascribed to a “mixed valence” effects (s. caption of Fig. 4 [now Fig. 5]) rather than to numerical accuracy. Do they have some evidence for making this statement?

Indeed we do. The figure included below, computed for our 3-band model system, shows the occupation probability of all energetically accessible atomic eigenstates as functions of temperature, where the collective quantum number $Q = (q, S, q_1 q_2)$ specifies the charge q , spin S and orbital SU(3) representation ($q_1 q_2$), labeled by a Young diagram with $q_1 + q_2$ (q_2) boxes in its first (second) row. Solid lines are for the Hund system ($U = 2, J = 1$), dashed lines for the Mott system ($U = 5.5, J = 1$). For both systems, the state $Q = (-1, 1, 01)$ (grey), with charge $q = -1$ (one below half filling), spin 1 and orbital degree of freedom in the fundamental SU(3) representation, has by far the largest occupancy. For the Mott system (dashed grey) it dominates completely, with an occupation probability of almost 0.9 at $T \approx 0.2$, where $T\chi_{\text{spin}}$ almost reaches 2/3 in Fig.5(e). At higher temperatures, other states become more important and the probability of $Q = (-1; 1; 01)$ decreases towards 0.7. By contrast, for the Hund system, the occupancy of the state $Q = (-1, 1, 01)$ is around 0.5 (solid grey), and states with other charges, $q = -2$ (orange) and $q = 0$ (dark red), have appreciable occupancies, too. This indicates that mixed-valence effects are not entirely negligible.

Reply to Reviewer #1

Reviewer #1 remains unpersuaded by our arguments emphasizing the differences between Mott and Hund metals. First, when discussing the liquid/gas example which we had raised in our previous reply, he/she argues: these examples are misleading in my opinion because they concern cases where there are zones of the phase diagrams where the changes are sudden between the two phases (a phase transition, or a sharp crossover). Correspondingly the differences between the two phases are very sharp in that region. This is in my opinion what justifies the distinction. In other parts of the phase diagram where the adiabatic connection results in a very smooth evolution of the physical properties typically this distinction is dropped.

This is a matter of perspective. Consider two points, A and B , on either side of the liquid/gas phase transition line. To tune the system from A to B , one can either cross the phase transition line, or avoid it by passing beyond the critical point where that line terminates. In the former case, the physical properties change abruptly via a phase transition, in the latter adiabatically via a crossover. In both cases, though, the properties at A and B differ dramatically. — From our perspective, drawing a distinction between systems with dramatically different properties is meaningful even if they are connected adiabatically. (While in this paper we illustrated a crossover between Hund and Mott systems, first order phase transitions between them might occur in other situations.)

Reviewer #1 then summarizes our work as follows: What the authors have done in this work, in my opinion, is taking two systems, both with correlations driven by \$U\$ and (most importantly) by \$J\$, but one close to the *density-driven* (in their response the authors talk about “doped Mott insulators” but here both systems have \$\langle N \rangle = 2\$ ) Mott transition and the other far from it. (We would like to clarify that we do not consider the density-driven MIT. The phrasing “doped Mott insulator” used (only once) in our previous reply was an inadvertent mistake. The MIT is driven by \$U\$ at fixed \$n_d = 2\$.) Both could be called “Hund metals”, and show the remarkable features (like the spin-orbital separation in the screening at low temperatures) of such metals compared to normal ones. Each one displays these features to a different degree, according to the different typical temperature scale of each.

On top of that, one of the two shows also the signs of the nearby interaction-driven Mott transition (which can be identified with the authors' definition of large E_{atomic} compared to the screening scale). It thus appears unnecessary to contrast Hund and Mott metals, as hereby defined.

To summarize, Reviewer #1 takes the view that since both materials studied by us have a non-zero Hund's coupling, both are Hund metals, differing only in their distance to an interaction-driven Mott transition. Since Hund and Mott metals are adiabatically connected in our model by varying the parameter U , this is a perfectly reasonable perspective. On the other hand, in our view it is just as reasonable to emphasize the fact that increasing U pushes the onset scales $T_{\text{orb}}^{\text{onset}}$ and $T_{\text{spin}}^{\text{onset}}$ closer together until they essentially coincide (see the new Fig. 1, added upon suggestion of Reviewer #3), thereby causing dramatic changes in the behavior of physical quantities (e.g. the temperature dependence of the susceptibilities). Therefore we believe, in contrast to Reviewer #1, that it *is* useful and meaningful to draw a distinction between materials deep in the Hund or Mott regimes, based on the differences in the temperature dependence of the screening process. This is clearly not a matter of right or wrong, but of perspective, with reasonable arguments available for both points of view. Reviewer #3, it appears, shares our's.

To do justice to both points of view, we have phrased our discussion of Fig. 1 (bottom half of page 2) in such a manner that it addresses both the adiabatic connection of Hund and Mott regimes and the dramatic differences in their physical behavior: “The most striking observation is that increasing U pushes the onset scales $T_{\text{orb}}^{\text{onset}}$ and $T_{\text{spin}}^{\text{onset}}$ closer together until they essentially coincide. As a consequence, the Hund regime (small U) and the Mott regime (large U , close to the Mott transition), though adiabatically connected via a crossover regime, show dramatic differences for the temperature dependence of physical quantities (discussed below).”

Returning to the liquid gas analogy, another reason why it is useful is because there are actual systems which clearly fit into each category. There are fluids where the spacing between the molecules or atoms is comparable to the size of the molecules or atoms themselves (prototypical liquids). And there are fluids where the inter-constituent spacing is much larger than their size (prototypical gases). For our system, the analogous quantity is the size of the region showing Hund-metal behavior (where orbitals and spins behave markedly different, with clear manifestations for their correlation functions, as discussed in our paper). We demonstrate this quantitative distinction for two important materials – in the ruthenate the Hund region is considerably larger than in the vanadate – thus making our model discussion very relevant to the field of correlated electron materials.

REVIEWERS' COMMENTS:

Reviewer #3 (Remarks to the Author):

I have read with interest the revised manuscript by Xiaoyu Deng et al., "Signatures of Mottness and Hundness in archetypical correlated metals", as well as their detailed reply to the Referee's criticisms.

Overall, I find that the Authors made a good job in answering to my observations and suggestions.

The additional calculations for $J=0$, the inclusion of the new general scheme of Fig. 1, as well as the modifications to the presentation of the numerical results have improved the general readability of the manuscript, making the message it aims to convey stronger. This was an important aspect, because the classification of materials proposed by the Authors in this work represents -in itself- a delicate issue, due to its intrinsic "crossover"-like nature.

I think, therefore, that the revised manuscript is now better suited to be considered for publication in Nature Communications. Prior to my final recommendation, the Authors should nonetheless consider the (relatively minor) points listed below, and make corresponding changes in their final manuscript, where they believe that it is necessary.

*) Fig. 1: I think that this schematic figure (and some of its keys/legends) should be enlarged to make it more readable. Furthermore, I wonder whether the white band between the grey-striped intermediate region and the full grey-colored "Mott" region does have a specific meaning (e.g., is this related to a change of slope of $T_{\text{onset}}^{\text{orb}}$?)

If not, I would suggest to continue the oblique grey lines of the intermediate regime up to the grey area of the Mott region, in order to avoid possible misunderstandings.

*) The new $J=0$ calculation: A relatively important point is the following. It is well known that, by considering the $J=0$ case with the same value of U , the impact of correlations changes considerably (e.g., one gets a relevant increase of U_{MIT} for the perfectly half-filled configuration, etc.). For this reason, I do not think that the Author should underline particularly this aspect in their discussion of the $J=0$ results (e.g., as in the added paragraph at the end of p.2 of the Supplemental Material).

Rather, the essential point for me is whether, for $J=0$, a separation of spin-orbital onset scales is absent **independently** of the actual correlation degree associated to the parameter set considered. If this general feature, as I understand, can be already inferred from the Author's new calculations for $J=0$, I would emphasize it more when discussing the comparison w.r.t. the $J=1$ case. At the same time, I would mention the overall suppression of correlation effects when comparing data at the same U for different J just as a known (here somewhat side) aspect of the problem.

I believe that this point is important, because the separation of the (spin, orbital) onset scales is the criterion, which -according to the Author's work- emerges as the most promising for distinguishing among the Hund's and Mott regimes.

*) Related to this, I would suggest the Authors to better clarify the following point: In the Caption of Fig. 5, they claim that: "For M_0 in (e) and W_0 in (g), deviations from Curie behavior set in at similar temperatures for spin and orb. ". If I understand correctly, however, this coincidence of the "onset" scales are **not** explicitly shown in the Figure, because they are located at much higher T than those plotted. If so, it would be necessary to add a "[not shown]" after this sentence in the caption, and, at the same time, show the corresponding data explicitly somewhere else (e.g. in the Supplemental material).

*) Captions: some captions (as, e.g., the same Fig. 5 discussed above) have become quite long,

which may hinder the readability. While I am not 100% sure about the format prescriptions in Nature Communications, I wonder whether the Authors could try to render their caption more concise (without sacrificing the essential information), either by splitting the figure or moving some details back to the main text.

Further, I noted that in the Captions sometimes a reference to a "purple arrow" is made. In my version of the manuscript, however, I could see only orange arrows. Hence, please carefully check the correspondence between the color coding (and the overall figure legend) and what is actually shown in the Figures. As the latter are often full with data and information, even a slight mismatch in the figure keys can be deceiving for the readers.

) The Authors have included in their reply to my criticisms an interesting plot, in order to clarify my question about the high-T deviation of the Curie-Weiss "plateau" of $T^ \chi$ w.r.t. purely local moment estimates. As future readers may have the similar doubts, I would strongly suggest to include shortly (but explicitly) this information in the Supplemental Material, making reference to it in the point of the main text, where such deviations are mentioned.

Detailed response to Reviewer #3's comments

Color code: blue: Reviewer #3's comments; black: our reply; purple: revisions/additions to the paper. For ease of reference, the principal changes in the main text are also typeset purple in the resubmitted version. References and figure numbers mentioned below refer to the revised submission.

Reviewer #3:

*) Fig.1: I think that this schematic figure (and some of its keys/legends) should be enlarged to make it more readable. Furthermore, I wonder whether the white band between the grey-striped intermediate region and the full grey-colored "Mott" region does have a specific meaning (e.g., is this related to a change of slope of $T_{\text{orb}}^{\text{onset}}$?) If not, I would suggest to continue the oblique grey lines of the intermediate regime up to the grey area of the Mott region, in order to avoid possible misunderstandings.

We thank Reviewer #3 for her/his useful suggestions. We enlarged the schematic figure such that the figure legend size matches the figure caption size. The "white band" mentioned by Reviewer #3 might be a display problem for specific pdf viewers. There is no white band "BETWEEN the grey-striped intermediate region and the full grey-colored 'Mott' region". We checked this using Adobe Acrobat X and Illustrator. The white region to the left is the Hund region, the intermediate region with oblique grey lines is a crossover region from the Hund to the Mott region, and the grey region to the right is the Mott region. To clarify this point we added a sentence in the caption of Fig.1: **White (grey) background indicates Hund (Mott) behavior at small (large) Δ_b . A crossover between both (oblique grey lines) is found at intermediate Δ_b .**

*) The new $J = 0$ calculation: A relatively important points is the following. It is well known that, by considering the $J = 0$ case with the same value of U , the impact of correlations changes considerably (e.g., one gets a relevant increase of U_{MIT} for the perfectly half-filled configuration, etc.). For this reason, I do not think that the Author should underline particularly this aspect in their discussion of the $J = 0$ results (e.g., as in the added paragraph at the end of p.2 of the Supplemental Material). Rather, the essential point for me is whether, for $J = 0$, a separation of spin-orbital onset scales is absent *independently* of the actual correlation degree associated to the parameter set considered. If this general feature, as I understand, can be already inferred from the Author's new calculations for $J = 0$, I would emphasize it more when discussing the comparison w.r.t. the $J=1$ case. At the same time, I would mention the overall suppression of correlation effects when comparing data at the same U for different J just as a known (here somewhat side) aspect of the problem.

I believe that this point is important, because the separation of the (spin, orbital) onset scales is the criterion, which -according to the Author's work- emerges as the most promising for distinguishing among the Hund's and Mott regimes.

We agree with Reviewer #3 that the $J = 0$ case is a well-known case. We therefore compare our $J = 1$ results against this case - following the suggestion in the report of Reviewer #3 of November 12, 2018: "in order to directly visualize the difference between the two situations: metallic systems relatively far from the Mott MIT vs. Hund's metal." In the Supplementary material we summarize known facts: the difference in the mass renormalization between the $J = 0$ and $J = 1$ case and the absence of spin-orbital separation in frequency-dependent zero-temperature spectral data for $J = 0$. However, in the main paper, we present our new and essential insight (see caption of Fig.6): "Thus the onset of screen-

ing shows spin-orbital separation for H1, but not for M1 (due to its proximity to the Mott transition), and also not for M0 and W0 (since these have $J = 0$).” To emphasize this point even more we have now added an additional sentence in the main text: **This spin-orbital separation of onset scales is a decisive fingerprint of Hund systems: it is absent for M1 and, for $J = 0$, it is absent for both W0 and M0, i.e. independently of the degree of correlations.**

*) Related to this, I would suggest the Authors to better clarify the following point: In the Caption of Fig.5 (now Fig.6), they claim that: “For M0 in (e) and W0 in (g), deviations from Curie behavior set in at similar temperatures for spin and orb.” If I understand correctly, however, this coincidence of the “onset” scales are *not* explicitly shown in the Figure, because they are located at much higher T than those plotted. If so, it would be necessary to add a “[not shown]” after this sentence in the caption, and, at the same time, show the corresponding data explicitly somewhere else (e.g. in the Supplemental material).

For clarification we supplemented the sentence in the caption of Fig.6 “For M0 in (a) and W0 in (c), deviations from Curie behavior set in at similar temperatures for χ_{spin} and χ_{orb} since $\chi_{\text{spin}} = 1.5\chi_{\text{orb}}$ for $J = 0$ (M0: small brown double arrow, W0: outside temperatures range of plot).”

*) Captions: some captions (as, e.g., the same Fig. 5 discussed above) have become quite long, which may hinder the readability. While I am not 100% sure about the format prescriptions in Nature Communications, I wonder whether the Authors could try to render their caption more concise (without sacrificing the essential information), either by splitting the figure or moving some details back to the main text. Further, I noted that in the Captions sometimes a reference to a “purple arrow” is made. In my version of the manuscript, however, I could see only orange arrows. Hence, please carefully check the correspondence between the color coding (and the overall figure legend) and what is actually shown in the Figures. As the latter are often full with data and information, even a slight mismatch in the figure keys can be deceiving for the readers.

We followed the advise to split Fig.5 into two figures (now Fig.5 and Fig.6). We further changed “purple arrow” to “orange arrow” in the figure captions. This was indeed an error and we thank Reviewer #3 for this important remark.

) The Authors have included in their reply to my criticisms an interesting plot, in order to clarify my question about the high-T deviation of the Curie-Weiss “plateau” of T^χ w.r.t. purely local moment estimates. As future readers may have the similar doubts, I would strongly suggest to include shortly (but explicitly) this information in the Supplemental material, making reference to it in the point of the main text, where such deviations are mentioned.

We thank Reviewer #3 for this valuable suggestion. We have now created a new section in the Supplemental material entitled “Mixed-valence effects” which shows and discusses the occupation probability of all energetically accessible atomic eigenstates as functions of temperature for M1 and H1. A reference to this information is given in the caption of Fig.6: “(Deviations of the observed plateaus from these local moment values reflect admixtures of states with different occupancy or spin, see Supplementary Figure 3 [47].)”